# Deep Augmentation:
# Dropout as Augmentation for Self-Supervised Learning

**Rickard Brüel-Gabrielsson    Tongzhou Wang    Manel Baradad    Justin Solomon**
*Massachusetts Institute of Technology*
*{brg, tongzhou, manelbaradad, jsolomon}@mit.edu*

Reviewed on OpenReview: *https://openreview.net/forum?id=OjWB2671AR*

## Abstract

Despite dropout's ubiquity in machine learning, its effectiveness as a form of data augmentation remains under-explored. We address two key questions: (i) When is dropout effective as an augmentation strategy? (ii) Is dropout uniquely effective under these conditions? To explore these questions, we propose Deep Augmentation, a network- and modality-agnostic method that applies dropout or PCA transformations to targeted layers in neural networks. Through extensive experiments on contrastive learning tasks in NLP, computer vision, and graph learning, we find that uniformly applying dropout across layers does not consistently improve performance. Instead, dropout proves most beneficial in deeper layers and can be matched by alternative augmentations (e.g., PCA). We also show that a stop-gradient operation is critical for ensuring dropout functions effectively as an augmentation, and that performance trends invert when moving from contrastive tasks to supervised tasks. Our analysis suggests that Deep Augmentation helps mitigate inter-layer co-adaptation—a notable issue in self-supervised learning due to the absence of labeled data. Drawing on these insights, we outline a procedure for selecting the optimal augmentation layer and demonstrate that Deep Augmentation can outperform traditional input-level augmentations. This simple yet powerful approach can be seamlessly integrated into a wide range of architectures and modalities, yielding notable gains in both performance and generalization.

## 1 Introduction

Self-supervised learning has emerged as a powerful paradigm in machine learning, enabling the creation of representations and pre-trained models without reliance on human-annotated labels. It has propelled breakthroughs in computer vision (Chen et al., 2020), natural language processing (Devlin et al., 2019), graph learning (Zhu et al., 2021), speech processing (Oord et al., 2016), and genomics (Zaheer et al., 2020). Within this landscape, *contrastive learning* (Oord et al., 2018; Chen et al., 2020) has gained particular prominence by leveraging augmentations that generate complementary pairs of samples, thereby preserving semantic structure (Shorten & Khoshgoftaar, 2019) and effectively expanding the training set.

Despite recent progress, effective augmentation strategies in contrastive learning often hinge on domain-specific knowledge—for instance, cropping and blurring in images (Chen et al., 2020), and token masking in NLP (Gao et al., 2021). Designing such augmentations can be time-consuming and may not generalize well across diverse modalities. This motivates the exploration of more universally applicable techniques.

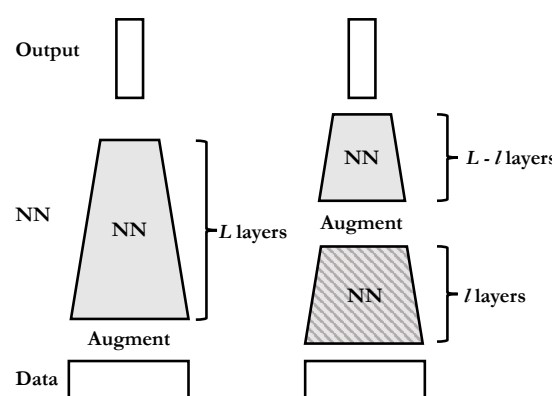

Figure 1: Left: Traditional augmentation. Right: Deep Augmentation at layer $l$.

A simple yet under-explored direction is to treat *dropout* (Labach et al., 2019; Salehin & Kang, 2023) not solely as a regularization tool, but also as a form of data augmentation (Srivastava et al., 2014). Dropout randomly zeros out activations, and its potential as an augmentation has been noted in self-supervised settings (Gao et al., 2021); however, the precise conditions under which it proves most effective remain unclear.

To address this gap, we introduce *Deep Augmentation* (Figure 1), a network- and modality-agnostic technique for augmenting high-dimensional activations in neural networks. Deep Augmentation applies dropout (Srivastava et al., 2014) or principal component analysis (PCA) (F.R.S., 1901) to specific layers, optionally combined with a stop-gradient operation. This design enables us to pose and investigate two central questions:

1. *When is dropout effective as an augmentation strategy?*
2. *Is dropout uniquely effective under these conditions?*

Through extensive experiments, we find that applying dropout uniformly across all layers does not consistently improve contrastive learning performance; instead, targeting deeper layers yields substantial gains. Moreover, we show that a stop-gradient operation is sometimes *critical* for harnessing the full potential of dropout-based augmentation, as it prevents detrimental gradient flow through the augmented representations. Notably, a similar effect can be achieved using PCA-based augmentations, suggesting that dropout is not uniquely suited to these conditions.

Interestingly, the performance trends reverse under supervised learning, indicating that Deep Augmentation mitigates inter-layer co-adaptation—a challenge that arises in the absence of labeled data. Our approach is straightforward to integrate into standard architectures such as ResNets (He et al., 2016), Transformers (Vaswani et al., 2017), and Graph Neural Networks (Kipf & Welling, 2017). It does not depend on manually designed augmentations or labeled datasets, making it an appealing option for a wide range of tasks.

**Contributions.** In summary, our main contributions are as follows:

- We show that dropout, when selectively applied to deeper layers (with or without a stop-gradient), can substantially improve performance in contrastive settings, while offering little benefit (or even proving detrimental) in supervised tasks.
- We demonstrate that dropout is not uniquely effective: a simple PCA-based augmentation can yield comparable gains.
- We highlight the significance of a stop-gradient operation for stabilizing and enhancing the effects of layer-targeted dropout, ensuring consistent performance boosts.
- We propose a practical procedure for selecting which layer to augment, showing that this choice is key to mitigating inter-layer co-adaptation.
- We thoroughly validate our approach on Transformers, ResNets, and Graph Neural Networks across multiple data modalities, underscoring the broad applicability of Deep Augmentation.

## 2 Related Work

**Self-Supervised Learning.** Self-supervised learning (Chen et al., 2020; Grill et al., 2020; Caron et al., 2020; Chen & He, 2021; Rani et al., 2023) leverages abundant unlabeled data to train transferable representations. By replacing human-annotated labels with automatically generated pseudo-labels, it enables training larger models on extensive datasets while mitigating overfitting. This paradigm has become increasingly popular for learning high-quality representations across diverse downstream tasks.

**Data Augmentation.** Data augmentation traditionally alleviates the scarcity of labeled data in supervised tasks, expanding the training set with semantically faithful transformations (Lecun et al., 1998). While contrastive and supervised learning often employ similar augmentations (Chen et al., 2020), label-dependent methods such as Mixup (Zhang et al., 2018a) are unique to supervised settings. In most self-supervised pipelines, augmentations focus on the *input space*, which, although intuitive, may not exploit the most semantically rich features available in deeper layers.

**Hierarchical and Higher-Layer Features.** Neural networks inherently learn hierarchical representations, with early layers capturing low-level patterns (e.g., edges) and deeper layers capturing higher-level semantics (e.g., textures, objects) (Bilal et al., 2018; Brüel Gabrielsson & Carlsson, 2019). Distances between higher-layer features can align with human judgments of semantic similarity (Zhang et al., 2018b; Ulyanov et al., 2018), underscoring their practical importance. Recent work has emphasized that capturing semantic granularity in deeper layers can notably enhance

downstream performance (Zhou et al., 2022; Hinton, 2023). Consequently, augmentations at these deeper layers can promote invariance to more abstract transformations than those found at the input level.

**Higher-Layer Augmentation.** Several approaches directly modify hidden or latent spaces. Manifold Mixup (Verma et al., 2018) applies mixup to hidden-layer outputs, and other work interpolates features for image classification (DeVries & Taylor, 2017). MODALS (Cheung & Yeung, 2021) unifies these ideas within a reinforcement-learning framework. Together, these studies highlight the potential of deeper-layer augmentation as a complement—or even alternative—to input-level transformations.

**Dropout as Augmentation.** *Dropout* (Labach et al., 2019; Salehin & Kang, 2023) is typically viewed as a regularizer that randomly zeroes out activations (Bouthillier et al., 2015), yet it can also serve as a general-purpose data augmentation (Srivastava et al., 2014). Recent work has explored this perspective in self-supervised contexts (Gao et al., 2021), but often by uniformly applying dropout across *all* layers. In contrast, we show that *selectively* targeting deeper layers can yield significant gains in contrastive learning, thereby challenging the one-size-fits-all assumption underlying uniform dropout usage. Our findings extend prior efforts that examine dropout's role in specialized settings (Wu & Gu, 2015), demonstrating its nuanced effects.

**Stop-Gradient & Information Collapse.** Stop-gradient mechanisms have been explored in Siamese networks (Chen & He, 2021), where they help avoid trivial collapses (Jing et al., 2022) and lessen the reliance on large batches or negative samples. In SimCLR (Chen et al., 2020), applying stop-gradient to one side of a positive pair *slightly* degraded performance (Chen & He, 2021), indicating that the effect may be context-specific. We broaden this investigation by introducing stop-gradient at different layers in conjunction with dropout- and PCA-based augmentations across diverse domains (including supervised tasks). From an information-theoretic perspective (Tian et al., 2020; Shwartz-Ziv & LeCun, 2023; Tishby & Zaslavsky, 2015), our layer-targeted augmentation appears to reduce unwanted uniformity or co-adaptation in latent features—an effect that can markedly improve contrastive learning performance while sometimes inhibiting performance on supervised tasks.

**Analysis of Representation Learning.** Prior theoretical works provide benchmarks for evaluating representation quality. For instance, Wang & Isola (2020) propose two criteria for contrastive-learning representations: *alignment* of positive pairs and *uniformity* on the hypersphere. Kornblith et al. (2019) introduce a similarity index (equivalent to centered kernel alignment, CKA) for comparing representational similarity across neural networks. We draw on these and similar methods to analyze how our Deep Augmentation strategy shapes the internal representations.

## 3 Method

### 3.1 Preliminaries

Contrastive learning seeks to learn representations by drawing semantically similar pairs closer while pushing dissimilar pairs apart. Given a dataset $X = \{x_1, \ldots, x_N\}$, it forms pairs $\mathcal{D} = \{(x_i^1, x_i^2)\}_{i=1}^m$, where $x_i^1$ and $x_i^2$ are distinct views of the same underlying sample $x_i \in X$, yet semantically similar. Constructing such pairs is pivotal, as it defines the invariances captured by the learned representations. Commonly, pairs are generated by applying random transformations (e.g., cropping, flipping, distortion) to the same sample.

Formally, let $Z \sim \mu$ be a random variable, where $\mu$ is a probability distribution over some space $\Omega$. (For instance, $\Omega$ could be discrete—e.g., cropping size—or continuous—e.g., blurring variance.) Let $A : \mathbb{R}^d \times \Omega \to \mathbb{R}^d$ be an augmentation function, and let $B \subset X$ be a randomly drawn batch. For each sample $x_i \in B$, draw $z_i^1, z_i^2 \sim \mu$. The features of the augmented pairs are defined as

$$h_i^j := f_\theta(A(x_i, z_i^j)) \quad \text{for } j \in \{1, 2\},$$

where $f_\theta$ is a neural network with learnable parameters $\theta$. The InfoNCE loss (Chen et al., 2020) for batch $B$ is then:

$$l(\theta; B) \;=\; \frac{1}{|B|} \sum_{i=1}^{|B|} \log \frac{e^{\text{cosine-sim}(h_i^1, h_i^2)/\tau}}{\sum_{j=1}^{|B|} e^{\text{cosine-sim}(h_i^1, h_j^2)/\tau}}, \tag{1}$$

where $\tau$ is a temperature parameter. This loss encourages $f_\theta$ to be invariant to the augmentation $A$ and promotes a uniform distribution of normalized features (Wang & Isola, 2020).

### 3.2 Deep Augmentation

A neural network $f_\theta$ processes data by successively applying $L$ layers. For $-1 \le a \le b < L$, let $f_\theta^{a,b}$ denote the operation from layer $a$ to layer $b$, where $a = -1$ represents the input itself, and $f_\theta^{a,a}$ is the identity. In particular, $f_\theta = f_\theta^{l+1,L-1} \circ f_\theta^{-1,l}$ for any $-1 \le l < L$. For example, applying an input augmentation corresponds to:

$$f_\theta(A(x_i, z_i^j)) \;=\; f_\theta^{0,L-1}\big(A(f_\theta^{-1,-1}(x_i), z_i^j)\big).$$

In this work, we investigate:

$$g_\theta^l(x_i, z_i^j) \;=\; f_\theta^{l+1,L-1} \circ A\big(f_\theta^{-1,l}(x_i), z_i^j\big), \tag{2}$$

where $-1 \le l < L$, as illustrated in Figure 1. Throughout, our primary augmentation $A$ is dropout, but we also explore an alternative PCA-inspired augmentation (Section 5). For dropout, $z_i^j$ corresponds to a random mask that zeros out a specific percentage of activations in $x_i$, meaning that $z_i^j$ is the source of stochasticity in the augmentation. Ideally, $A$ should satisfy three properties:

1. *Layer-agnostic*—the same $A$ applies at any layer $l$ without modification.
2. *Network-agnostic*—$A$ does not depend on a specific architecture for $f_\theta$.
3. *Modality-agnostic*—$A$ does not depend on the input domain.

Both dropout and our PCA-based approach satisfy these criteria.

We aim to identify which value(s) of $l$ yield the best representation $g_\theta^l$, as measured by downstream performance. When combining Deep Augmentation with standard input-level transformations, we simply compose the two augmentations (i.e., first apply the input-space augmentation, then apply the in-network augmentation).

**PCA Augmentation.** To test whether dropout is uniquely effective in Deep Augmentation, we compare it with a *principal-component removal* variant. Given a mini-batch $I_b = \{1, \ldots, K\}$, define the mean $\mu = \frac{1}{K} \sum_{k \in I_b} x_k$, the centered samples $\tilde{x}_k = x_k - \mu$, and the stacked matrix $\widetilde{X} = [\tilde{x}_1, \ldots, \tilde{x}_K] \in \mathbb{R}^{d \times K}$, and compute the singular-value decomposition $\widetilde{X} = U\Sigma V^\top$. Denote by $V_p \in \mathbb{R}^{d \times p}$ the first $p$ right singular vectors (the top $p$ PCs). Each sample is augmented as

$$A_p(x_i) \;=\; \underbrace{\big(I - V_p V_p^\top\big)}_{\text{project away from top } p \text{ PCs}} \; (x_i - \mu) \;+\; \mu.$$

Thus we subtract the components of $x_i$ that lie in the subspace spanned by the batch's top $p$ principal directions, using the batch itself (not an external noise vector) to generate the perturbation. We refer to this augmentation as *PCA* in subsequent sections.

**Stop-Gradient.** In equation 2, once $l > -1$, learnable layers exist *before* the augmentation. We optionally apply a stop-gradient operation at layer $l$, which prevents gradients from flowing into these earlier layers (Chen & He, 2021). This design enables us to distinguish two regimes: one where the network learns invariance only to the *already applied* augmentations (when using stop-gradient), and one where it additionally learns invariance to *upcoming* augmentations (when not using stop-gradient). In effect, by cutting off the gradient at the augmentation point, we prevent the upstream layers from "learning to undo" the perturbation, thereby preserving the desired invariances.

**Partial Batch Sampling.** By default, we apply Deep Augmentation to a random 50% of each mini-batch. This enhances variation in the training process and ensures that some samples remain unaugmented, maintaining alignment with evaluation conditions. Moreover, this approach preserves learning in all layers even if some samples experience a stop-gradient (it only applies to half of the batch). See Appendix A.5 for an ablation study.

**Co-Adaptation.** We define *co-adaptation* between layers as a scenario where two layers capture essentially the same information. To quantify this, we measure activation similarity using the centered kernel alignment (CKA) index (Kornblith et al., 2019). A high CKA value indicates that the two layers are effectively redundant. Since deterministic transformations cannot create new information (Shwartz-Ziv & LeCun, 2023), strong co-adaptation implies that minimal additional processing occurs in the deeper layer. Conversely, weaker co-adaptation suggests more effective information filtering, which can foster better generalization.

# 4 Main Results

We show that *Deep Augmentation* consistently and substantially improves contrastive learning performance across vision (Table 2), natural language processing (Table 1), and graph-based learning (Table 3). These gains primarily stem from the targeted use of dropout and stop-gradient; they need not reflect the absolute best performance achievable under every possible Deep Augmentation hyperparameter.

**Sentence Embeddings.** We follow the approach of Gao et al. (2021), pre-training a BERT transformer (Devlin et al., 2019) on $10^6$ randomly sampled sentences from English Wikipedia. Hyperparameters are tuned on the STS-B development set (Cer et al., 2017), and final evaluations are conducted on seven standard semantic textual similarity (STS) tasks (Agirre et al., 2012; Cer et al., 2017; Marelli et al., 2014).

**Vision.** For images, we employ a ResNet (He et al., 2016) and follow the SimCLR framework (Chen et al., 2020), testing on CIFAR10, CIFAR100, and a 100-class subset of ImageNet (Deng et al., 2009). Again, we target deeper layers for dropout and stop-gradient, leading to measurable improvements over standard SimCLR.

**Graph Contrastive Learning.** In graph-based tasks, we adopt the GCL framework (Zhu et al., 2021) with a GCN backbone (Kipf & Welling, 2017). We evaluate on COLLAB and IMBD-Multi (Yanardag & Vishwanathan, 2015), as well as NCI1 (Wale & Karypis, 2006) and PROTEINS (Borgwardt et al., 2005). Hyperparameters are tuned on a validation split, with results reported on a separate test set. Table 3 shows that applying Deep Augmentation in deeper layers benefits performance across most datasets.

**Compute & Memory Savings.** Stop-gradient can reduce both training time and memory usage. In our setup, the portion of the network cut off by stop-gradient no longer computes gradients, saving roughly $4\times$ the compute time and $3\times$ the memory for the affected layers. Most of these savings are realized on the GPU. By selectively applying stop-gradient to deeper layers and to only half of each batch, overall training time drops to about $62.5\%$ of the baseline, while memory usage is trimmed to about $66\%$. Tables 1, 2, and 3 provide approximate "Compute" metrics reflecting these savings.

Table 1: Contrastive Learning on Sentence Embeddings with Transformer. Performance on STS tasks (Spearman's correlation, where higher is better). SimCSE versus SimCSE with Deep Augmentation, specifically layer-targeted dropout and stop-gradient at layer 8. Compute refers to the estimated use of compute time and memory, as compared to SimCSE.

| Model | STS12 | STS13 | STS14 | STS15 | STS16 | STS-B | SICK-R | Avg. | Compute |
|---|---|---|---|---|---|---|---|---|---|
| SimCSE | $66.71_{\pm0.505}$ | $81.13_{\pm1.279}$ | $73.13_{\pm1.818}$ | $80.82_{\pm0.593}$ | $\mathbf{78.47}_{\pm0.644}$ | $77.54_{\pm0.906}$ | $71.49_{\pm0.904}$ | $75.61_{\pm0.924}$ | 100% |
| SimCSE+DeepAug | $\mathbf{69.00}_{\pm1.111}$ | $\mathbf{81.82}_{\pm0.127}$ | $\mathbf{74.48}_{\pm0.311}$ | $\mathbf{81.84}_{\pm0.439}$ | $78.41_{\pm0.146}$ | $\mathbf{78.63}_{\pm0.114}$ | $\mathbf{71.75}_{\pm0.442}$ | $\mathbf{76.56}_{\pm0.161}$ | ~79% |

Table 2: Contrastive Learning in Vision with ResNets. SimCLR versus SimCLR with Deep Augmentation, specifically layer-targeted dropout and stop-gradient at layer 4, across all datasets. Values represent classification accuracies (higher is better). Compute refers to the estimated use of compute time and memory, as compared to SimCLR.

| Model | CIFAR10 | CIFAR100 | ImageNet100 | Compute |
|---|---|---|---|---|
| SimCLR | 90.37 | 61.64 | 79.38 | 100% |
| SimCLR+DeepAug | **91.04** | **64.01** | 79.56 | ~66% |

Table 3: Contrastive Learning on Graphs with GNNs. GCL (Graph Contrastive Learning) versus GCL with Deep Augmentation, specifically layer-targeted dropout and stop-gradient at layer 6, across all datasets. Values represent classification accuracies (higher is better) measured in f1mi (Micro-averaged F1 Score). Compute refers to the estimated use of compute time and memory, as compared to GCL.

| Model | COLLAB | IMDB-Multi | NCI1 | PROTEINS | Compute |
|---|---|---|---|---|---|
| GCL | $72.40_{\pm0.6}$ | $\mathbf{53.33}_{\pm1.1}$ | $73.97_{\pm1.6}$ | $72.32_{\pm1.5}$ | 100% |
| GCL+DeepAug | $\mathbf{73.80}_{\pm1.3}$ | $52.89_{\pm4.2}$ | $\mathbf{75.83}_{\pm1.0}$ | $\mathbf{73.21}_{\pm1.5}$ | ~66% |

# 5 Ablations

Deep Augmentation introduces additional hyperparameters (e.g., the targeted layer, augmentation type, stop-gradient usage). In this section, we present ablation studies demonstrating its consistent performance across different datasets. We analyze the impact of (i) contrastive vs. supervised settings, (ii) layer depth, (iii) stop-gradient usage, (iv) dropout vs. PCA augmentations, and (v) pre-trained initialization.

## 5.1 Sentence Embeddings

SimCSE (Gao et al., 2021) first showed that using *only* dropout can improve contrastive learning for sentence embeddings (built on a pre-trained MLM model). We extend SimCSE's setup by applying layer-specific dropout or PCA, with or without stop-gradient, both *with* and *without* additional MLM augmentations.

**Augmentation, Layer, and Stop-Gradient.** Figures 2a and 2b illustrate our results using (a) dropout and (b) PCA-based transformations on the STS-B development set. For PCA, we tested removing both the largest and sixth-largest principal component; Figure 2b reports the largest component, which worked best. The sixth-component results appear in the Appendix.

Overall, Deep Augmentation surpasses plain SimCSE across a range of layer depths. Adding stop-gradient further boosts performance with dropout, though the gains are smaller for PCA.

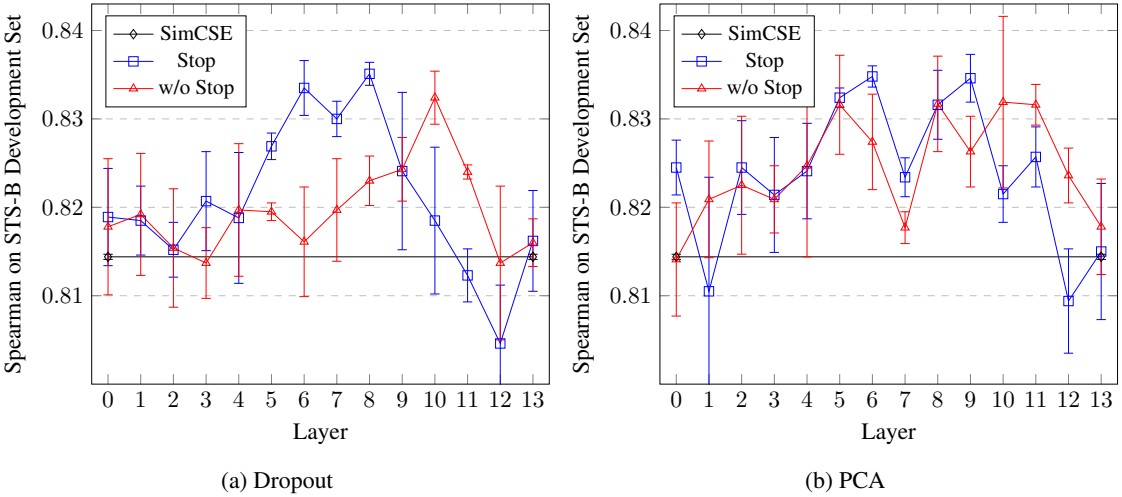

(a) Dropout                                         (b) PCA

Figure 2: SimCSE vs. Deep Augmentation with (a) Dropout or (b) PCA, with and without stop-gradient. "Stop": stop-gradient. Deep Augmentation outperforms SimCSE.

**Deep Augmentation and Masked Language Modeling (MLM).** Figure 3 compares Deep Augmentation to MLM's original data augmentations in SimCSE. Although SimCSE already tunes dropout rates (0%, 1%, 5%, 10%, 15%, 20%), our fixed 50% dropout rate at a chosen layer still yields higher results, underscoring the robustness of layer-targeted augmentation alongside MLM—this reduces dependece on pre-trained models and development sets, by supporting simultaneous Deep Augmentation and MLM training. Across standard STS tasks, the best Deep Augmentation setup exceeds SimCSE's strongest Spearman correlation (e.g., 74.32 vs. 69.31). Moreover, even in a purely MLM setting (i.e., no contrastive objective), Deep Augmentation significantly improves performance (Appendix, Figure 51), and there the effect of stop-gradient is less pronounced.

**Supervised Learning.** To contrast Deep Augmentation's impact in supervised vs. contrastive learning, we train on STS-B directly as a supervised task. Figures 4a and 4b show results with dropout and PCA, respectively, with/without stop-gradient. Contrary to our contrastive findings, Deep Augmentation *reduces* performance in the supervised setting—especially in higher layers. This stark difference supports our hypothesis that deeper-layer augmentations help most when no explicit supervision is provided, forcing the model to learn robust invariances on its own.

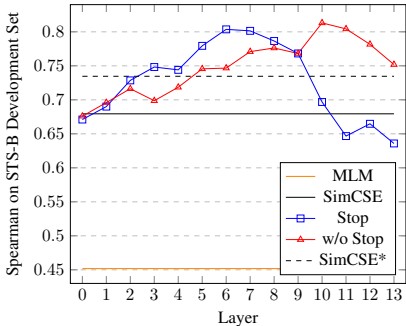

Figure 3: SimCSE vs. Deep Augmentation with and without stop-gradient, both with MLM. "Stop": stop-gradient. *: includes hyperparameter search over dropout rates.

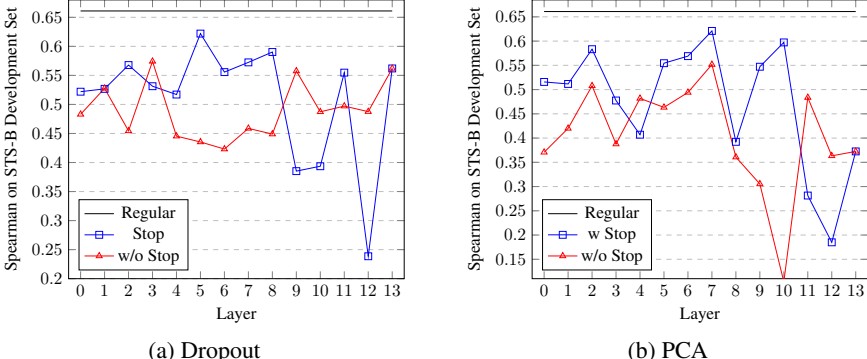

(a) Dropout                    (b) PCA

Figure 4: Supervised only. Deep Augmentation with (a) Dropout (best performing across dropout rates .5, .25, .125) and (b) PCA, with and witout stop-gradient, on STS-B.

## 5.2   Vision

For vision, our primary experiments use CIFAR-100 (Appendix reports CIFAR-10). ImageNet100 results follow the same layer-targeted dropout and stop-gradient configuration determined optimal for CIFAR.

**Architecture.** We adopt ResNet18 (Appendix A.1, Table 4). Compared to Transformers and GNNs, ResNet has less regularity across depths: ResNet contains convolutional layers upfront, followed by a fully connected layer, with average pooling interspersed. This, in addition to the varying dimensionality across layers, makes vision-specific hyperparameters more challenging.

**Dropout is Not Sufficient Alone.** Unlike for sentence embeddings, using *only* dropout as the augmentation did not yield competitive results in vision. Consequently, we complement deep augmentations with standard image augmentations. Future work could investigate adapting pre-trained vision models to new domains via dropout alone (similar to SimCSE in NLP).

**Data Augmentation and Targeted Dropout.** Figure 14, in Appendix, show that uniform dropout across all layers degrades contrastive learning, but layer-specific dropout can mitigate these losses. Notably, 50% dropout uniformly applied is detrimental, whereas targeting the same 50% rate to certain layers has much less impact on performance.

**Augmentation, Layer, and Stop-Gradient.** Now we apply Deep Augmentation with dropout (Figure 5a), with and without stop-gradient. Deep Augmentation with dropout and stop-gradient demonstrate significant performance improvements, particularly for layers 4 and 6; however, not using the stop-gradient did not achieve performance comparable to using stop-gradient. A small tuning of dropout rate yielded the results in Table 2 (Appendix A). We also evaluate Deep Augmentation with PCA augmentation, removing the largest and the sixth largest principal component. Removing the sixth largest yields superior performance (results with and without stop-gradient in Figure 5b). For the largest, see the Appendix. Similar to dropout, stop-gradient consistently enhances performance, especially in higher layers.

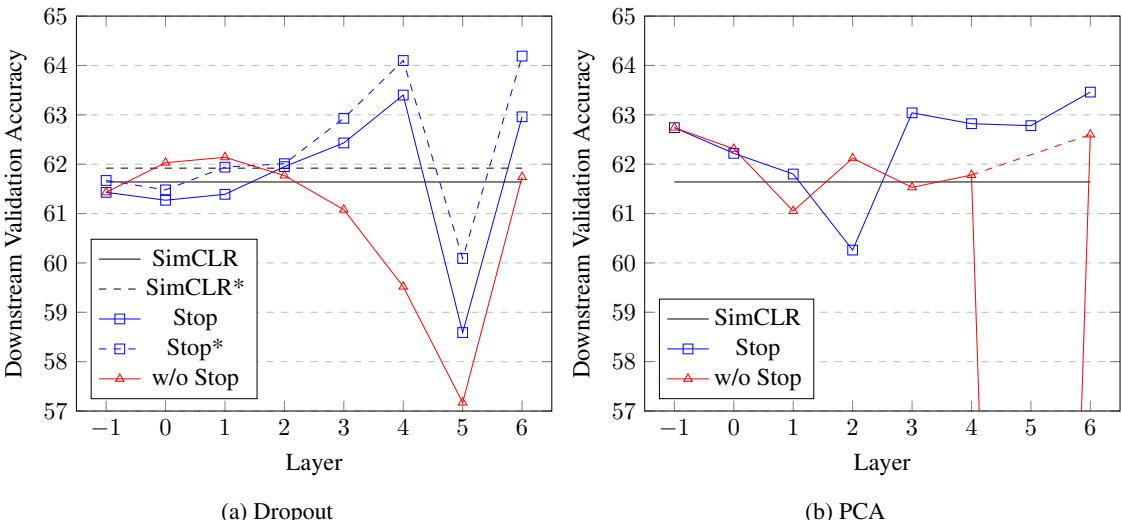

(a) Dropout

(b) PCA

Figure 5: Contrastive learning. Deep Augmentation with (a) dropout or (b) PCA, with and without stop-gradient. *: initialized with pre-trained SimCLR model. "Stop" is short for stop-gradient. Note: Layer 5 is an average pooling.

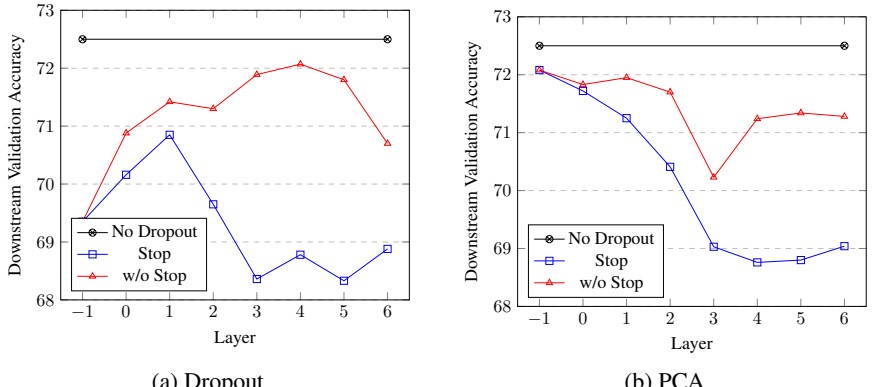

(a) Dropout

(b) PCA

Figure 6: Supervised only. Deep Augmentation with (a) dropout or (b) PCA, with and without stop-gradient. *: initialized with pre-trained SimCLR model. "Stop" is short for stop-gradient.

**Initialization & Freezing Weights.** Augmentations may have a more significant impact on higher layers that already possess useful, discriminative features. In addition, the concurrent objectives of learning features and maintaining invariance to their alterations could conflict, slowing down or destabilizing training. In Figure 5a, 'SimCLR*' and 'Stop*' apply Deep Augmentation atop a SimCLR-pretrained ResNet. While using a pre-trained initialization outperforms not using a pre-trained initialization, the gains are marginal, indicating that extensive pre-training is not essential for Deep Augmentation. Similarly, freezing specific layers before or after the augmentation point does not yield notable benefits (Appendix A.7).

**Supervised Learning.** Figures 6a and 6b compare Deep Augmentation's effect in supervised vision tasks. Here, dropout or PCA does not provide benefits. Notably, omitting stop-gradient actually performs better—opposite to our contrastive results. Retaining basic data augmentations remains crucial; removing them lowers accuracy to around 59.02%.

## 5.3 Graphs

Finally, we evaluate Deep Augmentation on graph contrastive learning, extending our insights from vision and NLP to GNNs. We use standard Graph Contrastive Learning (GCL) augmentations (Zhu et al., 2021) on COLLAB, IMDB-Multi, NCI1, and PROTEINS. In GCL, a graph is augmented—using operations such as node and edge deletion—to

create two distinct views, and the model is trained to identify pairs of views originating from the same graph. We use the f1mi (Micro-averaged F1 Score) metric; further details can be found in Appendix C.

**Augmentation, Layer, and Stop-Gradient.** Figures 7 and 8 show results for dropout and PCA (removing the sixth-largest principal component). Since graph inputs vary in embedding size per number of nodes, we adapt PCA to operate over across node embeddings in the batch. While trends vary somewhat by dataset, Deep Augmentation *with* dropout and stop-gradient significantly boosts performance in most cases (especially at higher layers).

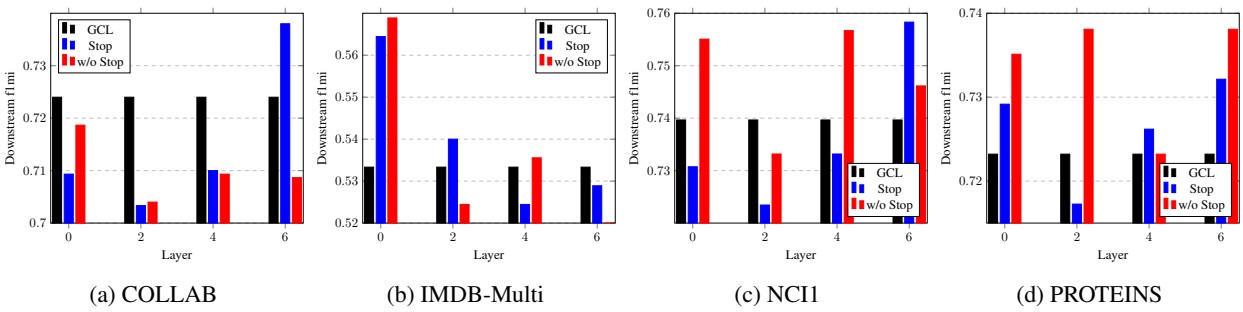

|(a) COLLAB|(b) IMDB-Multi|(c) NCI1|(d) PROTEINS|

Figure 7: Graphs: Deep Augmentation with Dropout

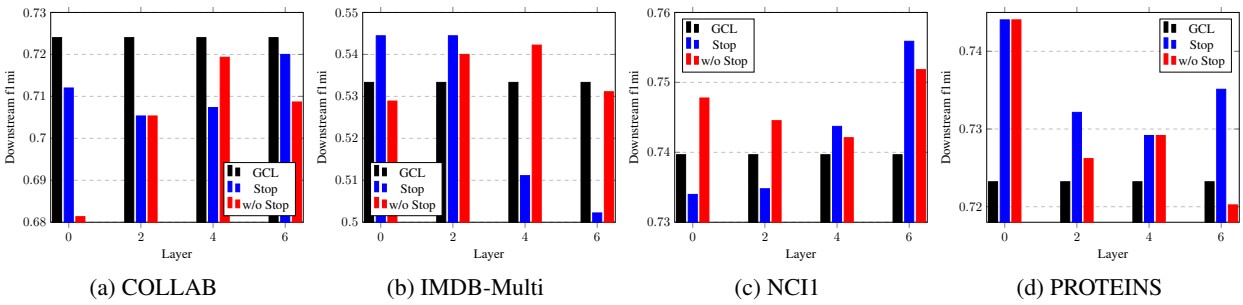

|(a) COLLAB|(b) IMDB-Multi|(c) NCI1|(d) PROTEINS|

Figure 8: Graphs: Deep Augmentation with PCA

## 5.4 Findings

In summary, our ablation studies indicate:

- Contrastive vs. Supervised Learning: Deep Augmentation generally has opposite effects on self-supervised versus supervised learning. See Figure 9. It is surprising that Deep Augmentation reduces performance in the supervised setting, particularly in higher layers, given that dropout is commonly effective in supervised learning. A likely explanation is that our Deep Augmentation setting departs from conventional dropout: we use larger dropout rates, confine the mask to a single layer, and adjust other hyperparameters. Accordingly, we emphasize the *direction* of the effect rather than the absolute numbers.
- Layer Depth: Applying augmentation to higher layers typically yields the largest gains in contrastive learning.
- Stop-Gradient: Improves contrastive performance across diverse data but often reduces accuracy in supervised tasks.
- Augmentation Type: Both dropout and PCA are effective, though they exhibit different behaviors and trade-offs.
- Pre-trained Weights: Starting from pre-trained models is not essential for successful dropout-based augmentation.

## 6 Analysis

Deep Augmentation demonstrates strikingly different effects in contrastive and supervised learning. This section explores *why* Deep Augmentation benefits contrastive learning while offering little or no gains in supervised contexts, and *how* we can determine which layers to target for best results. We integrate alignment–uniformity metrics (Wang & Isola, 2020) and CKA similarity (Kornblith et al., 2019) to show how these transformations reshape latent representations and mitigate co-adaptation among layers.

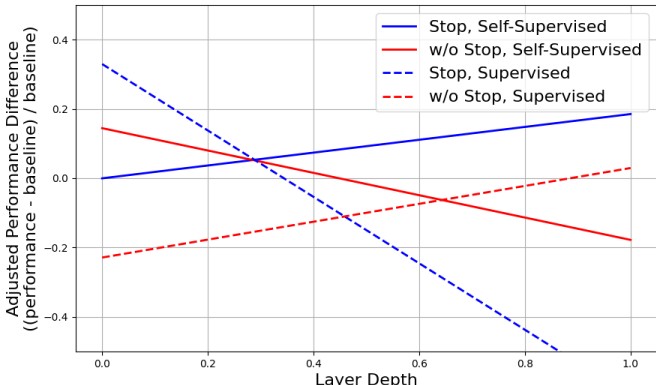

Figure 9: Regression lines are fitted to experimental results across NLP, vision, and graph-based tasks. All data have been $z$-score normalized (with the original mean preserved). In the self-supervised setting, deep augmentation improves performance in higher layers when using stop-gradient, while the opposite trend is observed without stop-gradient. In contrast, for supervised learning, deep augmentation generally does not yield benefits, and the trends across layers are reversed.

**Contrastive Learning.** Our findings indicate that Deep Augmentation helps reduce overfitting and eliminate spurious alignment in contrastive tasks, while also *maintaining or enhancing uniformity* across learned features. This is evident from alignment–uniformity analyses, which show improved invariance properties, and from CKA measures, which confirm that stronger feature transformations in targeted layers reduce inter-layer similarity (co-adaptation). Importantly, these techniques also identify *which layers* are most susceptible to co-adaptation, guiding the strategic selection of layers where Deep Augmentation is most beneficial.

**Supervised Learning.** In stark contrast, supervised learning sees little benefit—and can even suffer—when Deep Augmentation is applied. Because labeled tasks already specify ground-truth invariances (the intra-class differences), they do not rely on broad augmentations to combat spurious alignments. In effect, the network is already "regularized" by the labels themselves, leading to naturally lower co-adaptation in deeper layers. Consequently, additional perturbations introduced by Deep Augmentation can degrade performance. This aligns with studies on information bottlenecks (Tishby & Zaslavsky, 2015; Shwartz-Ziv & LeCun, 2023; Jing et al., 2022), which highlight how supervised training inherently enforces a more constrained representation space compared to contrastive methods optimizing mutual information across potentially infinite augmentations.

Taken together, these findings illuminate how Deep Augmentation simultaneously combats overfitting and enforces useful invariances in contrastive tasks, yet exerts little positive influence—and sometimes a negative one—when the target invariances are already determined by label supervision.

## 6.1  Co-Adaptation Between Layers

**Sentence Embeddings and Transformers.** Figure 10 shows CKA similarity for a Transformer model under different conditions:

- BERT: The pre-trained baseline for SimCSE and Deep Augmentation.
- SimCSE: Contrastive learning applied to BERT, notably reducing co-adaptation across its higher layers (i.e., 10–12).
- Layer 10 (w/o Stop): Deep Augmentation applied at Layer 10 (highlighted by a red cross) without stop-gradient.
- Layer 8 (w/ Stop): Deep Augmentation with stop-gradient at Layer 8 (highlighted by a red cross).

In BERT, layers 8–11 show a stretch of co-adaptation (between the black crosses). Deep Augmentation at or near these points reduces similarity in subsequent layers, overcoming the co-adaptation that persists even after SimCSE. Interestingly, choosing the earlier "cross" for stop-gradient and the later "cross" for without stop-gradient perform the best. This suggests a potential extension: targeting multiple layers could further boost performance. Overall, CKA

similarity index highlights a co-adaptation issue between layers and determines at what layer Deep Augmentation should be applied.

Appendix Figure 59 shows that, in a supervised setting, co-adaptation is already weaker even without Deep Augmentation, emphasizing that ground-truth labels inherently curb excessive co-adaptation. Further details are provided in Appendix E.

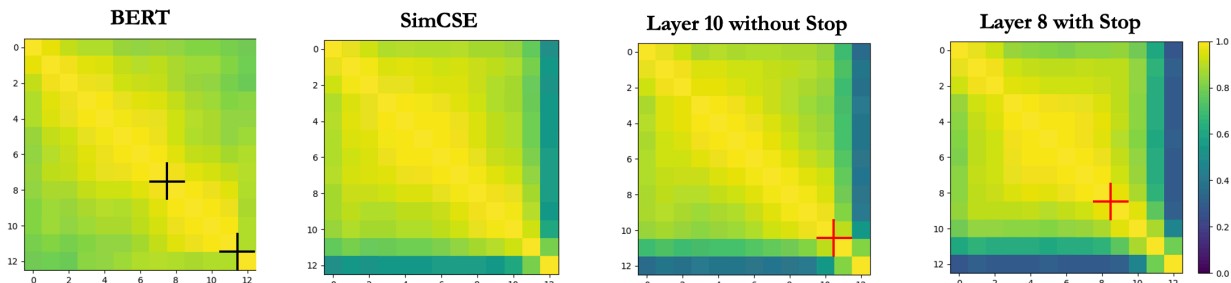

Figure 10: CKA similarity index for "BERT," "SimCSE," "Layer 10 without Stop" (Deep Augmentation applied without stop-gradient at Layer 10), and "Layer 8 with Stop" (Deep Augmentation applied with stop-gradient at Layer 8) on the STS-B dataset. Black crosses mark the beginning and end of a co-adaptation region in BERT, while red crosses on "Layer 10 without Stop" and "Layer 8 with Stop" highlight the targets of Deep Augmentation. Optimal performance of Deep Augmentation is observed near the black crosses, indicating its effectiveness and guiding the selection of layers for targeting.

**Images and ResNet.** Figure 11 compares CKA similarity for:

- Random: A randomly initialized ResNet18,
- SimCLR: ResNet18 trained with SimCLR,
- Layer 4 without Stop: ResNet18 with Deep Augmentation at Layer 4 (w/o Stop).

We find that co-adaptation between Layers 4 and 5 emerges after SimCLR training and is even stronger in the poorer-performing "Layer 4 (w/o Stop)." Conversely, the best-performing models (Layer 4 or Layer 6 *with* stop-gradient) avoid this excessive similarity. This corroborates the Transformer and sentence embeddings findings: certain layers (e.g., Layer 4 in ResNet18) are especially prone to co-adaptation, pinpointing where Deep Augmentation can be most beneficial. See Appendix E for more details.

Appendix Figure 56 shows a similar pattern to Transformer and sentence embeddings in supervised training, with consistently lower co-adaptation than in self-supervised settings (particularly in the last layer)—further evidence that labeled tasks intrinsically limit over-adaptation between layers.

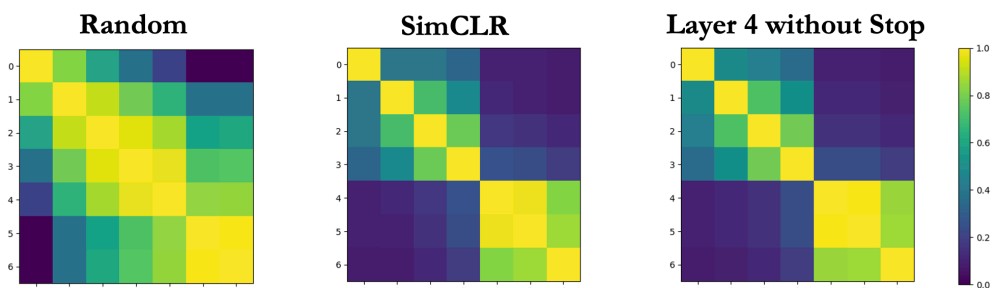

Figure 11: Indications of why Layer 4 is special in Figure 5a, as the major divide between co-adaptation across layers. Layers 0-4 are convolutional. All high-performing NNs have the pattern of SimCLR, and failure cases have stronger co-adaptation between Layers 4 and 5. "Layer 4 without Stop" corresponds to the failure case of Deep Augmentation without stop-gradient at Layer 4.

**Sentence Embeddings and Transformer.** Alignment and Uniformity measures for sentence embedding methods are in Figure 12, computed as in SimCSE (Gao et al., 2021) w.r.t. ground truth (STS-B development set), during training, and with methods converging to higher density regions.

Following Gao et al. (2021), we measure alignment and uniformity on the STS-B development set (Figure 12), during training, and with methods converging to higher density regions. Without MLM, Deep Augmentation trends toward better uniformity (lower is better) than SimCSE alone. Adding stop-gradient further enhances alignment at some expense of uniformity ("S" vs. "w/o S"). However, introducing MLM reverses the optimization direction, boosting alignment at some cost to uniformity. Again, the Deep Augmentation variants (*with* or *without* stop-gradient) tend to outperform the SimCSE baselines.

In Appendix Table E, we compare alignment and uniformity in supervised learning. Deep Augmentation slightly improves alignment but not uniformity, consistent with the notion that supervised models already have high uniformity on ground-truth labels, leaving less room for improvement.

Note, since these measures are computed on ground truth validation classes rather than augmentations, they offer a different perspective from the original introduction of these measures for contrastive learning Wang & Isola (2020).

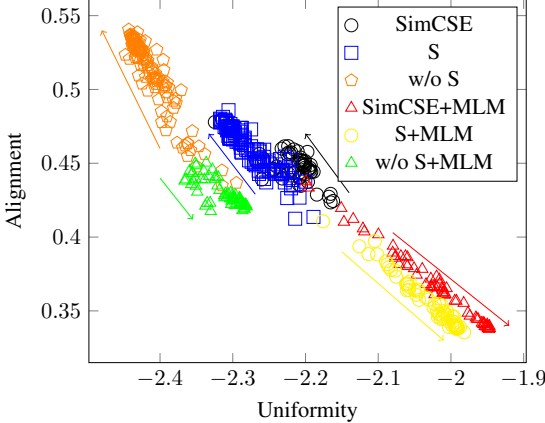

Figure 12: Alignment and Uniformity (lower is better) for sentence embeddings on STS-B: SimCSE vs. Deep Augmentation (with and without stop-gradient). We also include these methods combined with the pre-training method of BERT, i.e., Masked Language Modeling (MLM). Arrows indicate the direction during training, which reverses when MLM is introduced. "S" is short for stop-gradient.

**Images and ResNet.** For vision, we standardize on SimCLR augmentations to measure alignment and uniformity (Figure 13). Checking multiple training checkpoints (e.g., epochs 300, 600, 900, 1200, 1500), we observe that:

- Higher training epochs improve uniformity on test data and alignment on training data.
- Layers 4 and 6 (w/ Stop) consistently outperform SimCLR on test alignment/uniformity, avoiding the overfitting to alignment that SimCLR shows on training data.
- Layer 4 (w/o Stop) matches SimCLR on these metrics but underperforms on downstream tasks, highlighting these metrics' limitations in capturing latent space quality. Hence, our preference for the CKA similarity index for comprehensive assessment.

Appendix Figure 54 shows that, in supervised training, the differences are less pronounced. However, Layer 4 (w/ Stop) perform better on both metrics but this does not translate to better classification accuracy. This highlights that the invariances in self-supervised learning and supervised learning remain fundamentally different, and superior performance on the self-supervised task does not necessarily translate to improved performance on the supervised downstream task. Avoiding overfitting to alignment is crucial when that alignment differs from the ground truth, but less critical when alignment matches the ground truth. Again, tasks with ground-truth labels do not see the same overfitting challenges as contrastive setups, reducing the need for strong regularization from Deep Augmentation.

# 7 Discussion

Our work affirms that dropout, widely known as a regularizer, can indeed serve as an effective *in-network* augmentation method—particularly within self-supervised pipelines where the risk of inter-layer co-adaptation runs high. The success of PCA-inspired augmentations further underscores that dropout is but one avenue for improving contrastive training in

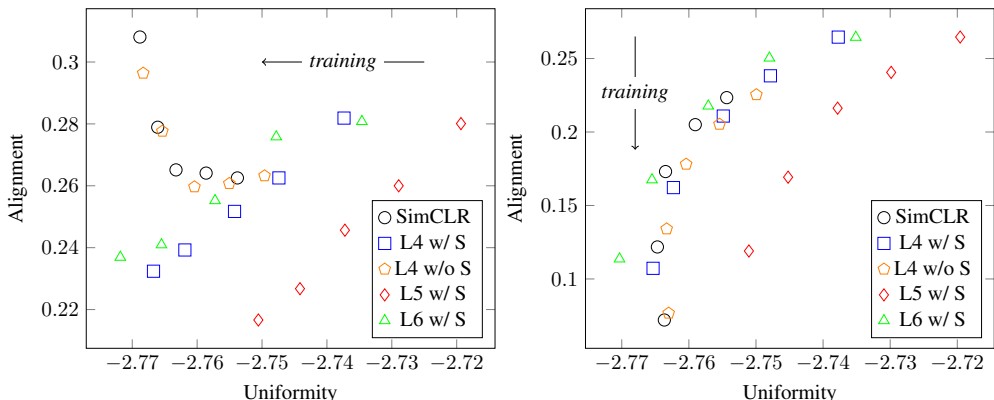

Figure 13: Alignment and Uniformity (lower is better) of SimCLR augmentations on CIFAR. Left: Test data. Right: Training data. Deep Augmentation outperforms SimCLR when measuring alignment and uniformity *using SimCLR's augmentations* on the test set, and SimCLR overfits at Alignment on the training set. "L" is short for Layer and "S" is short for stop-gradient.

deep networks. By providing a procedure for layer selection and underscoring the importance of stop-gradient, we hope this work encourages more granular thinking about dropout-like augmentations in deep learning, inspiring new ways to exploit activation-space transformations for both performance and generalization gains.

## 8 Acknowledgements

The MIT Geometric Data Processing Group acknowledges the generous support of Army Research Office grants W911NF2010168 and W911NF2110293, of National Science Foundation grant IIS2335492, from the CSAIL Future of Data program, from the MIT–IBM Watson AI Laboratory, from the Wistron Corporation, and from the Toyota–CSAIL Joint Research Center.

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

# A  CIFAR

In this section, we outline the training specifics for our experiments on the CIFAR datasets, complemented by supplementary results and comparative analyses.

## A.1  ResNet Architecture

The specifications for ResNet18 are detailed in Table 4.

Table 4: Configuration of ResNet18 on CIFAR

| ResNet18 on CIFAR | | |
| --- | --- | --- |
| Layer | Type | #Neurons |
| -1 | Input Data | $32^2 \times 3 = 3072$ |
| 0 | Conv(k=3, s=1) | $32^2 \times 64 = 65536$ |
| 1 | Conv(k=3, s=2) | $32^2 \times 64 = 65536$ |
| 2 | Conv(k=3, s=2) | $16^2 \times 128 = 32768$ |
| 3 | Conv(k=3, s=2) | $8^2 \times 256 = 16384$ |
| 4 | Conv(k=3, s=2) | $4^2 \times 512 = 8192$ |
| 5 | Avgpool | 512 |
| 6 | MLP | 128 |

## A.2  Dropout at All Layers Versus 50% Layer Targeted Dropout

In Figure 14, we compare dropout rates at all layers versus 50% dropout rate targeted at a specific layer.

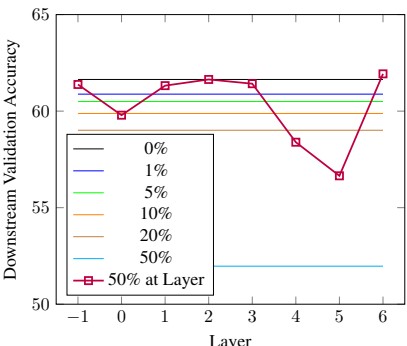

Figure 14: Comparing dropout rates at all layers versus 50% dropout rate targeted at a specific layer. For ratio of dropped to total nodes when targeting a layer, see Appendix A.3; there is no trend.

## A.3  Proportion of Dropped Nodes

When setting a dropout rate for a specific layer, it exclusively affects that layer. Consequently, a 50% dropout rate at one layer results in fewer neurons being dropped compared to a 50% dropout applied uniformly across all layers. Additionally, the same dropout rate can impact different numbers of neurons in various layers, reflecting the varying neuron counts in each layer.

In Table 5 we include the number of nodes in each layer, the total nodes across all layers. Thus, for $0.5$ dropout, we show the proportion of dropped nodes when a layer is targeted. There is not trend between the proportion and performance.

## A.4  Training Details

For implementation, we utilized the code provided by (Khosla et al., 2020), available at this link. Our experiments were conducted with a batch size of 1024, training each method for 1500 epochs.

Table 5: Proportion of Dropped Nodes per Layer at 50% dropout

| Layer | Dropped Nodes | Total Nodes | Proportion |
|-------|---------------|-------------|------------|
| 0 | $0.5 \times 65536$ | 192640 | 0.17 |
| 1 | $0.5 \times 65536$ | 192640 | 0.17 |
| 2 | $0.5 \times 32768$ | 192640 | 0.085 |
| 3 | $0.5 \times 16384$ | 192640 | 0.043 |
| 4 | $0.5 \times 8192$ | 192640 | 0.021 |
| 5 | $0.5 \times 512$ | 192640 | 0.001 |
| 6 | $0.5 \times 128$ | 192640 | 0.0003 |

## A.5 Naïve Deep Augmentation with stop-gradient on CIFAR100

In Figure 15, we include results of 50% dropout with stop-gradient at individual layers on 100% of the batch. Such naïve augmentations generally give poor performances. All layers besides the input data layer lead to downstream accuracy of 1% (equivalent with random guess). The input data layer arrives at a downstream accuracy of 61.38%.

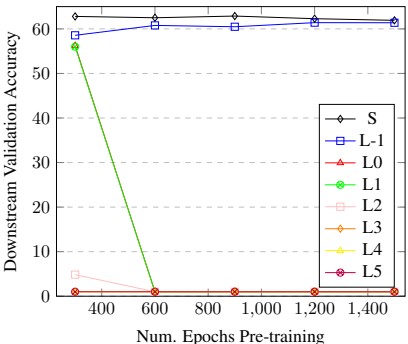

Figure 15: CIFAR100. 50% dropout with stop-gradient applied at individual layers on 100% of the batch. I.e. freezing earlier layers to random weights.

## A.6 Including Deep Augmentation w/o stop gradient initialized with SimCLR

For completion, we also include Deep Augmentation without stop gradient, initialized with pre-trained SimCLR model, together with the other variants—see Figure 16.

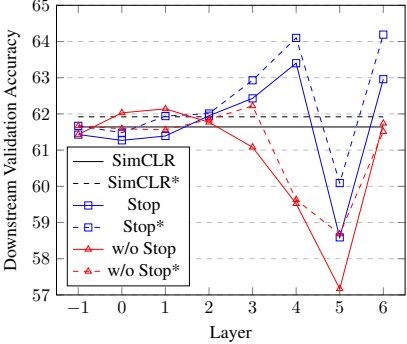

Figure 16: Comparing sampling 50% and applying 50% dropout, with or without stop-gradient. *: initialized with pre-trained SimCLR model. Stop: short for stop-gradient.

## A.7 Freezing

We repeat the experiment with pre-trained initialization but freeze all the layers up to and including the layer at which the targeted transformation occurs; see "Freeze before" in Figure 17. Compared to not freezing, this strategy gives very different results. In particular, the downstream performance of Layers 3 and 4 is critically reduced.

Deep Augmentation after frozen SimCLR layers may not work well due to co-adaptation between neurons, leading to overfitting. Suppose a layer of a NN exhibits strong co-adaptation within several subsets of neurons, i.e., each subset encodes a single data feature. Randomly dropping neurons is unlikely to remove a complete co-adapted subset of neurons. Ideally, features are learned per neuron so dropping any of them provides a complementary view. Alternatively, features might be represented continuously among neurons in a layer such that dropout corresponds to something akin to blurring the feature continuously.

Because early layers have fewer parameters to distort the input data, such layers may have less co-adaptation. This might explain why earlier layers, rather than later layers, perform better when frozen during Deep Augmentation. Similarly, higher layers may benefit from higher dropout rates because they are more susceptible to co-adaptation, explaining why in Figure 5a, Deep Augmentation in higher layers yields the best downstream performance.

Reversely, we may freeze the layers following the targeted layer; results are labeled "Freeze after" in Figure 17. Compared to "Freeze before", Layer 3 improves, Layer 5 worsens, while Layer 4 performs similarly. This asserts that later layers, some more than others, benefit from learning to be invariant to Deep Augmentation.

We see that Deep Augmentation with freezing layers and initialized to SimCLR-model, works better for earlier layers than for later layers. Especially in Figure 39a and 43, we see that earlier layers outperform SimCLR earlier in the training. This suggests that incrementally freezing layers, and adding Deep Augmentation at the next layer, might help improve performance and speed up training.

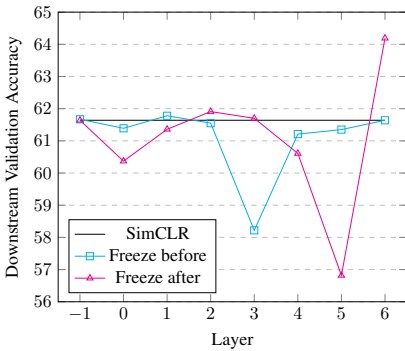

Figure 17: Freezing layers before and after Deep Augmentation with stop-gradient, initialized with pre-trained SimCLR model. For "Freeze before," Layer -1 freezes nothing, and for "Freeze after" Layer 6 freezes nothing.

## A.8 PCA Augmentation

In Figure 21, results demonstrate that removing the largest principal component from a batch sample is less effective than subtracting the sixth largest (Figure 28).

Figure 20 presents the six largest principal values from the layers of a randomly initialized ResNet18 versus one trained with SimCLR on CIFAR100. Post-SimCLR training, the distribution of values becomes more uniform, and there is a notable shift in the rank of layers before and after the training process.

## A.9 Supervised Learning

For our supervised learning experiments, training was conducted for 100 epochs but otherwise using the same hyperparameters as those in the fine-tuning phase post pre-training, which lasted 28 epochs.

Figure 23 presents results from supervised learning on CIFAR100, comparing the effects of uniform dropout across all layers with 50% dropout applied to a targeted layer.

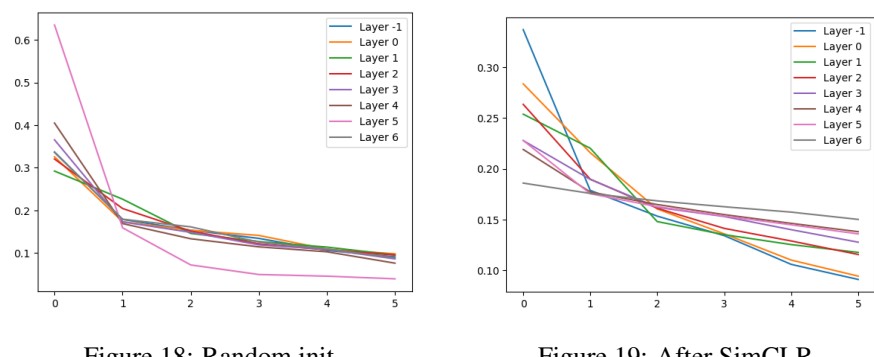

Figure 18: Random init.  Figure 19: After SimCLR

Figure 20: The six largest principal value from the layers of a random initialized ResNet18 and one trained with SimCLR on CIFAR100.

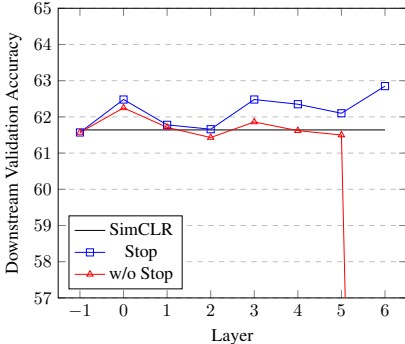

Figure 21: PCA: Comparing sampling 50% of batch and subtracting the largest principal component from that sample, with and without stop-gradient.

Figure 22 presents the results of supervised training but also includes standard deviations.

## A.10  Domain Transfer: CIFAR100 to CIFAR10

We perform basic domain transfer experiments by taking networks pretrained on CIFAR100 and finetuning them on CIFAR10. In Figure 24 we include results comparing SimCLR with Deep Augmentation with and without stop-gradient, across layers. We also include performance for different checkpoints across training, see Figure 25a and Figure 25b for Deep Augmentation with and without stop gradient, respectively. Note the overfitting tendencies.

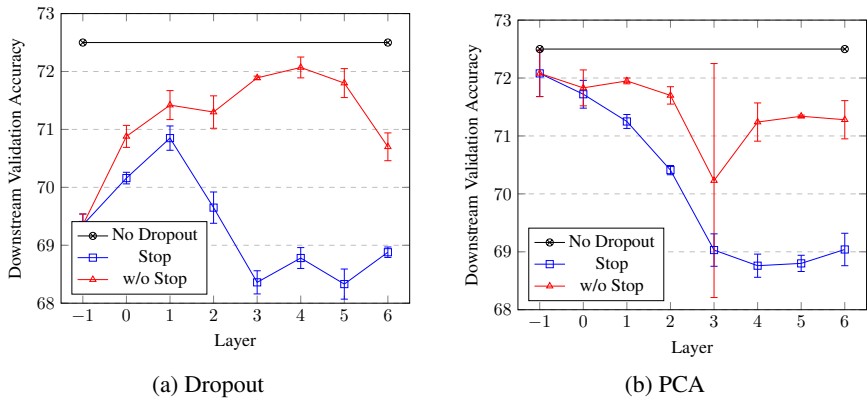

(a) Dropout  (b) PCA

Figure 22: Supervised only. Deep Augmentation with (a) dropout or (b) PCA, with and without stop-gradient. *: initialized with pre-trained SimCLR model. "Stop" is short for stop-gradient.

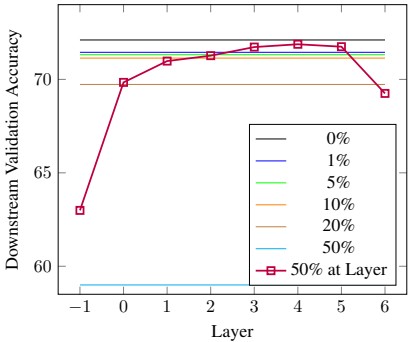

Figure 23: Supervised on CIFAR100: Comparing dropout rates at all layers versus 50% dropout rate targeted at a specific layer.

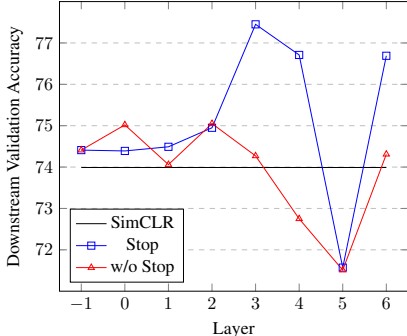

Figure 24: Finetuning on CIFAR10 of networks pre-trained on CIFAR100. Comparing SimCLR with Deep Augmentation with and without stop-gradient. Stop: short for stop-gradient.

## A.11 Different dropout rates

In Figure 26, we tune over dropout rates 0.5, 0.25, and 0.125 and find that 0.125 at Layer 4 performs the best.

## A.12 CIFAR10

We include results on most of the experiments that were run on CIFAR100, also on CIFAR10. In general, results show the same trends as for CIFAR100. In Figure 30, we include results comparing dropout rates across all layers to 50%

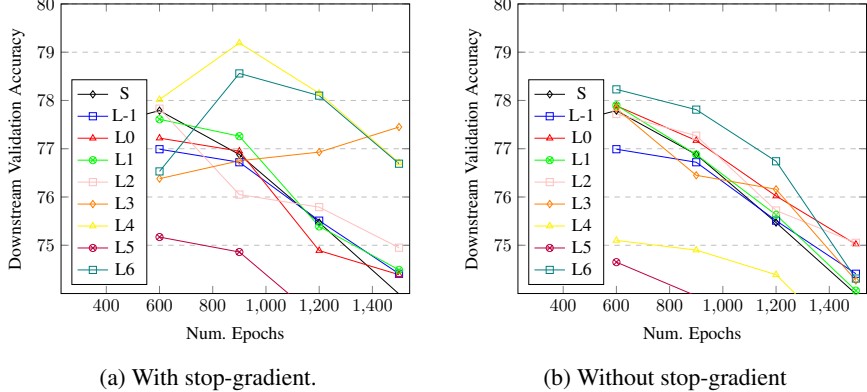

(a) With stop-gradient.  (b) Without stop-gradient

Figure 25: SimCLR and Deep Augmenation with and without stop-gradient pre-trained on CIFAR100 and finetuned on CIFAR10, for different checkpoints during training. Observe the overfitting behavior.

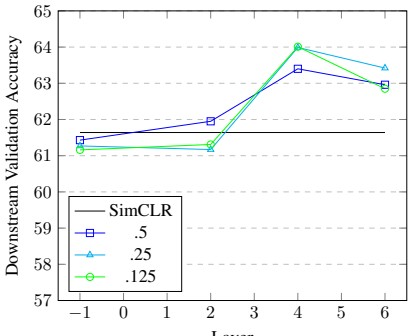

Figure 26: CIFAR100: Comparing sampling 50% of batch and applying dropout to that sample, with and without stop-gradient, for different dropout rates. "Stop" is short for stop-gradient.

dropout at single layers. Again, we see targeted dropout at some layers showing much better performance than dropout across all layers.

In Figure 31 we include results of sampling 50% of batch and performing 50% dropout with and without stop-gradient, called "Stop" and "w/o Stop" respectively. We also include a benchmark of SimCLR. Here "*" refers to the networks being initialized with a pre-trained SimCLR model. Again, we see Layer 4 (with stop-gradient) and Layer 6 (with and without stop-gradient) stand out. It is also interesting to note that when initializing with a pre-trained SimCLR model, performance differs significantly more for Deep Augmentation with stop-gradient than without.

In Figures 27 and 28, we subtract the largest and sixth largest principal component from half the samples of the batch.

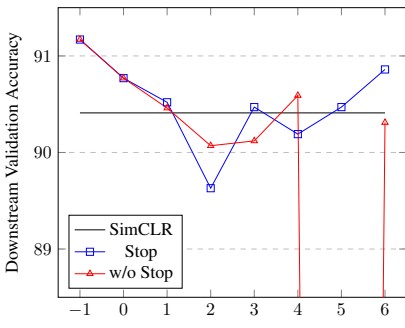

Figure 27: PCA CIFAR10: Comparing sampling 50% of batch and subtracting the largest principal component from that sample, with and without stop-gradient.

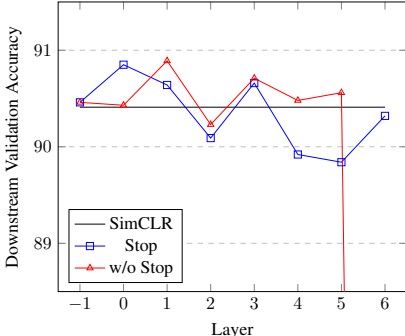

Figure 28: PCA CIFAR10: Comparing sampling 50% of batch and subtracting the sixth principal component from that sample, with and without stop-gradient.

In Figure 32, we include results of Deep Augmentation with stop-gradient but freezing layers up to the targeted layer versus freezing after the targeted layer. Again, we see the performance change, especially Layer 3 and 4 degrading, while Layer 2 improves.

In Figure 29, we include results of supervised learning on CIFAR10, with dropout across all layers as well as 50% dropout at targeted-layer.

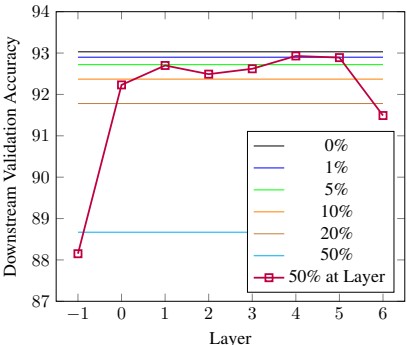

Figure 29: Supervised on CIFAR10: Comparing dropout rates at all layers versus 50% dropout rate targeted at a specific layer.

We perform basic domain transfer experiments by taking networks pretrained on CIFAR10 and finetuning them on CIFAR100. In Figure 33 we include results comparing SimCLR with Deep Augmentation with and without stop-gradient, across layers. We also include performance for different checkpoints across training, see Figure 45a and Figure 45b for Deep Augmentation with and without stop gradient, respectively. Note the overfitting tendencies.

In Figure 34, we tune over dropout rates 0.5, 0.25, and 0.125 and fine that 0.125 at Layer 4 performs the best.

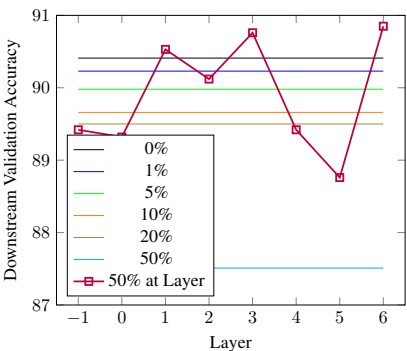

Figure 30: CIFAR10: Comparing dropout rates at all layers versus 50% dropout targeted at a specific layer.

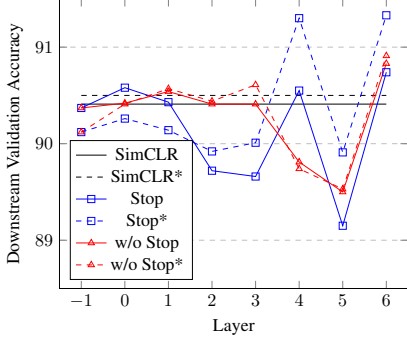

Figure 31: CIFAR10: Comparing SimCLR with Deep Augmentation with and without stop-gradient. *: Initialized with pre-trained SimCLR model. Stop: short for stop-gradient.

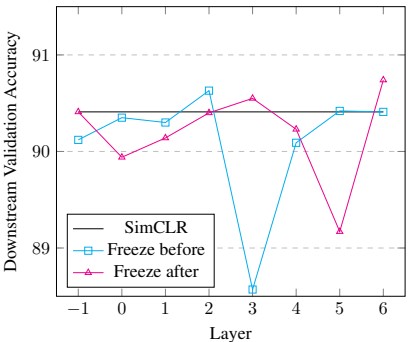

Figure 32: CIFAR10: Comparing freezing layers before or after Deep Augmentation with stop-gradient, initialized with pre-trained SimCLR model. Note that for "Freeze before" Layer -1 freezes nothing, and for "Freeze after" Layer 6 freezes nothing.

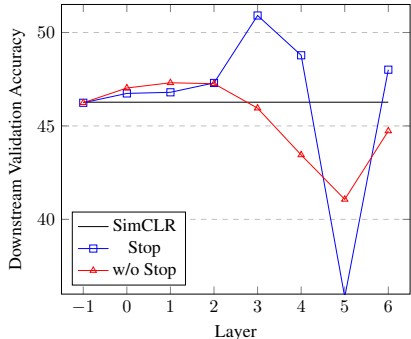

Figure 33: Finetuning on CIFAR100 of networks pre-trained on CIFAR10. Comparing SimCLR with Deep Augmentation with and without stop-gradient.

## A.13   CIFAR100 across epochs

We include results where we finetuned and tested checkpoints at different epochs for various experiments.

In Figure 35, we include results for dropout everywhere at different rates and 50% dropout at single layers.

In Figure 36, we include results for sampling 50% of each batch and performing 50% dropout on that sample, with and without stop-gradient.

In Figures 37 and 38, we include results for sampling 50% of each batch and subtracting the largest and sixth largest (respectively) principal component from that sample, with and without stop-gradient.

In Figure 39, we compare freezing layers before or after Deep Augmentation with stop-gradient initialized with pre-trained SimCLR model.

In Figure 40, we inlcude results for 50% sampling, 50% dropout, with and without stop-gradient, and initialized with pre-trained SimCLR model.

## A.14   CIFAR10 across epochs

We include results where we finetuned and tested checkpoints at different epochs for various experiments.

In Figure 41, we include results for dropout everywhere at different rates and 50% dropout at single layers.

In Figure 42, we include results for sampling 50% of each batch and performing 50% dropout on that sample, with and without stop-gradient.

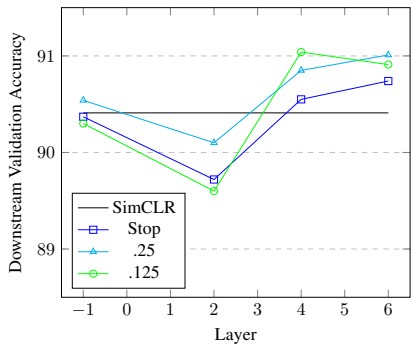

Figure 34: CIFAR10: Comparing sampling 50% of batch and applying dropout to that sample, with and without stop-gradient, for different dropout rates. "Stop" is short for stop-gradient.

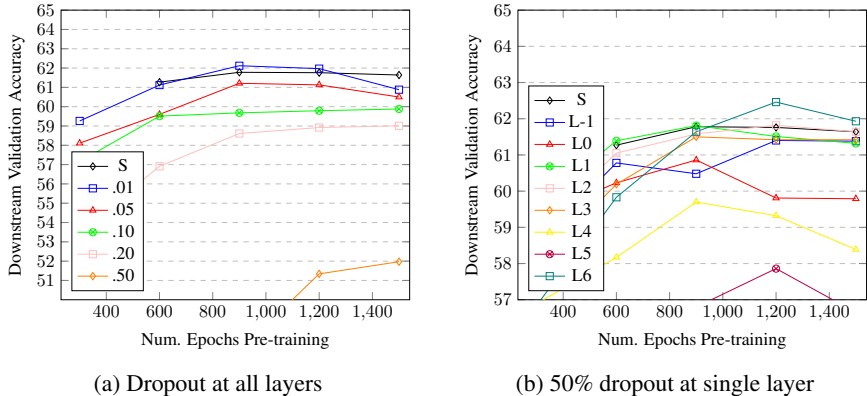

(a) Dropout at all layers

(b) 50% dropout at single layer

Figure 35: CIFAR100. Comparing dropout rates at all layers versus 50% dropout targeted at a specific layer. Note difference in $y$-axis.

In Figure 43, we compare freezing layers before or after Deep Augmentation with stop-gradient initialized with pre-trained SimCLR model.

In Figure 44, we inlcude results for 50% sampling, 50% dropout, with and without stop-gradient, and initialized with pre-trained SimCLR model.

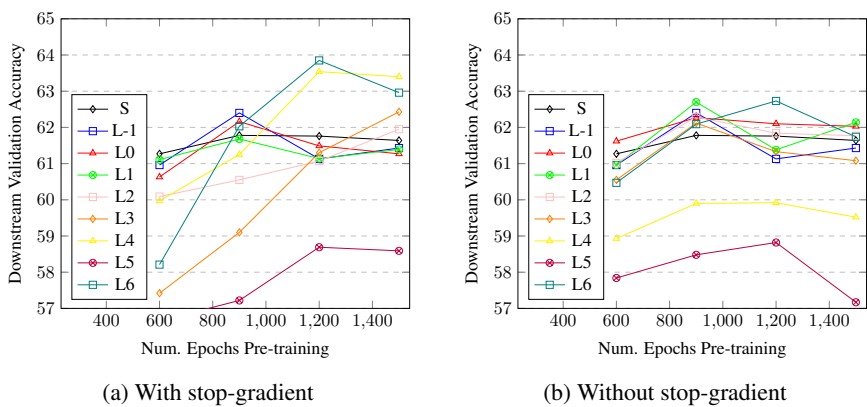

(a) With stop-gradient

(b) Without stop-gradient

Figure 36: CIFAR100. Comparing sampling 50% and applying 50% dropout, with or without stop-gradient.

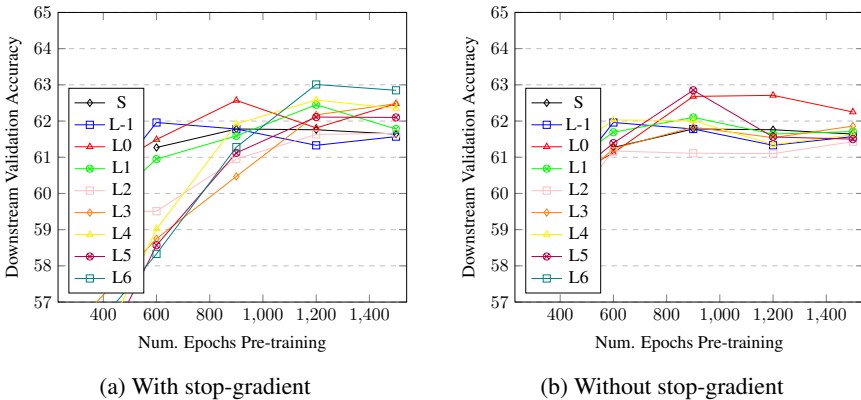

(a) With stop-gradient

(b) Without stop-gradient

Figure 37: PCA CIFAR100: Comparing sampling 50% of batch and subtracting the largest principal component from that sample, with and without stop-gradient.

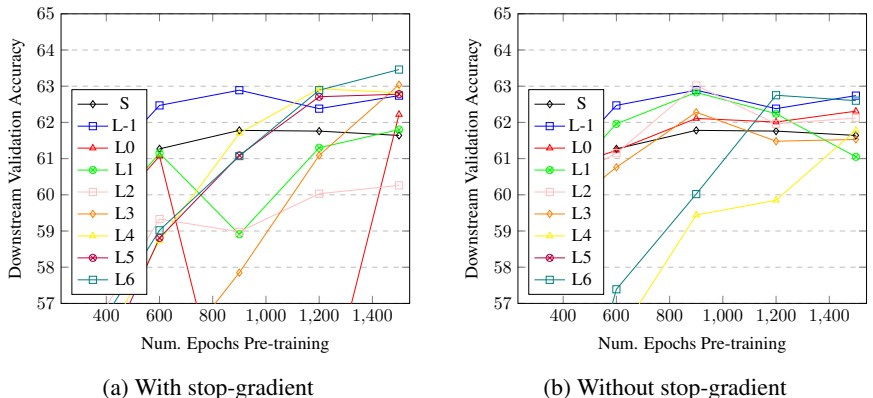

(a) With stop-gradient

(b) Without stop-gradient

Figure 38: PCA CIFAR100: Comparing sampling 50% of batch and subtracting the sixth largest principal component from that sample, with and without stop-gradient.

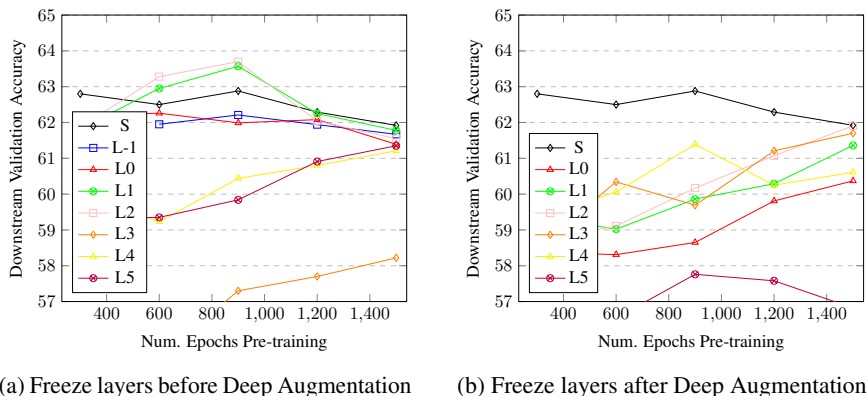

(a) Freeze layers before Deep Augmentation

(b) Freeze layers after Deep Augmentation

Figure 39: CIFAR100. Comparing freezing layers before or after Deep Augmentation with stop-gradient initialized with pre-trained SimCLR model.

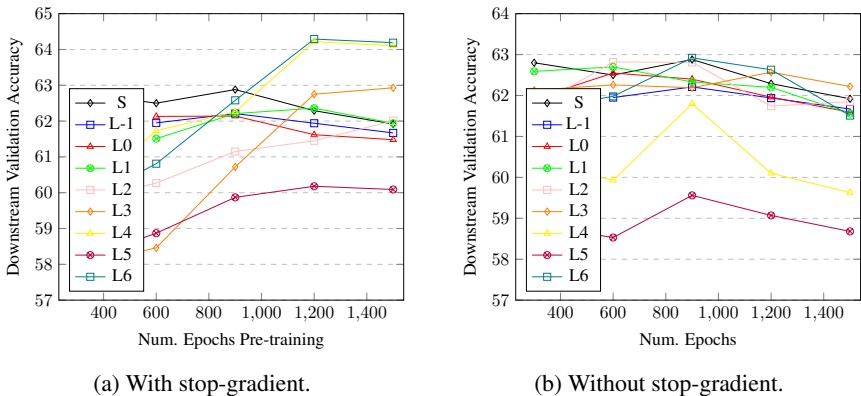

(a) With stop-gradient.

(b) Without stop-gradient.

Figure 40: CIFAR100. 50% sampling, 50% dropout, with and without stop-gradient, and initialized with pre-trained SimCLR model.

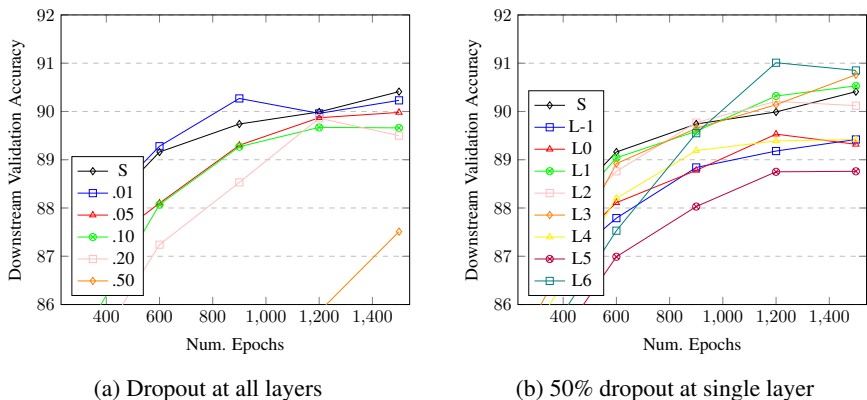

(a) Dropout at all layers

(b) 50% dropout at single layer

Figure 41: CIFAR10. Comparing dropout rates at all layers versus 50% dropout targeted at a specific layer. Note difference in $y$-axis.

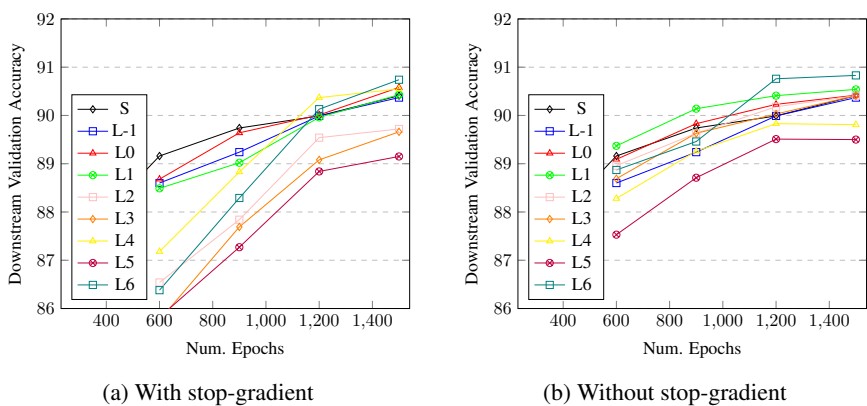

(a) With stop-gradient

(b) Without stop-gradient

Figure 42: CIFAR10. Comparing sampling 50% and applying 50% dropout, with or without stop-gradient.

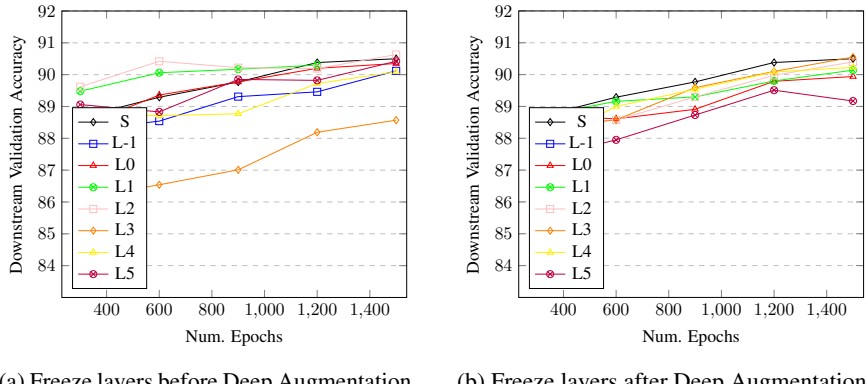

(a) Freeze layers before Deep Augmentation    (b) Freeze layers after Deep Augmentation

Figure 43: CIFAR10. Comparing freezing layers before or after Deep Augmentation with stop-gradient initialized with pre-trained SimCLR model.

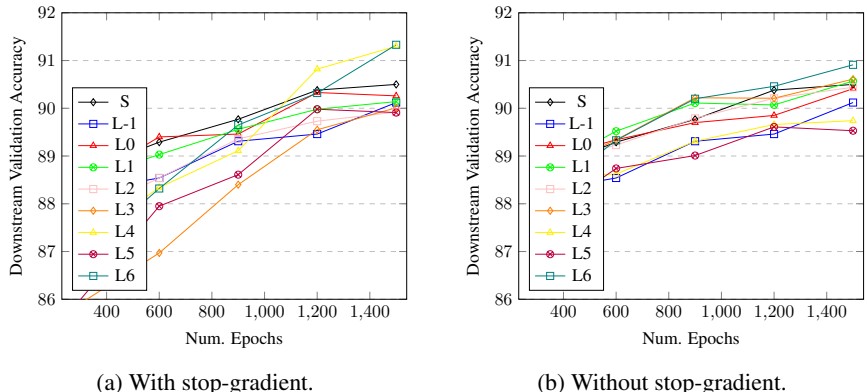

(a) With stop-gradient.    (b) Without stop-gradient.

Figure 44: CIFAR10. 50% sampling, 50% dropout, with and without stop-gradient, and initialized with pre-trained SimCLR model.

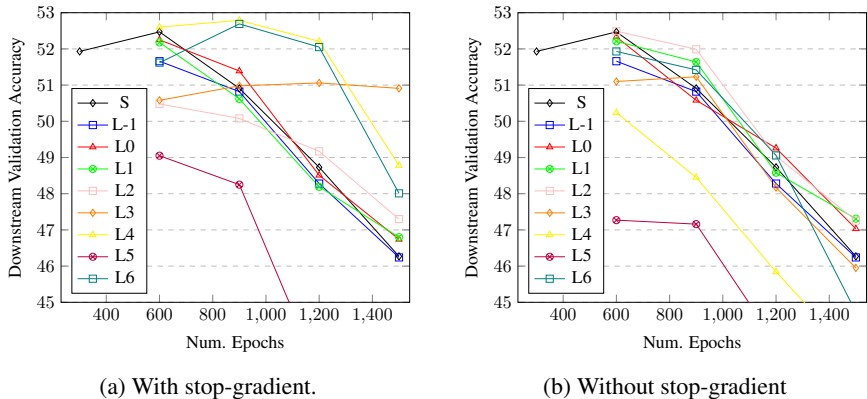

(a) With stop-gradient.    (b) Without stop-gradient

Figure 45: SimCLR and Deep Augmenation with and without stop-gradient pre-trained on CIFAR10 and finetuned on CIFAR100, for different checkpoints during training. Observe the overfitting behavior.

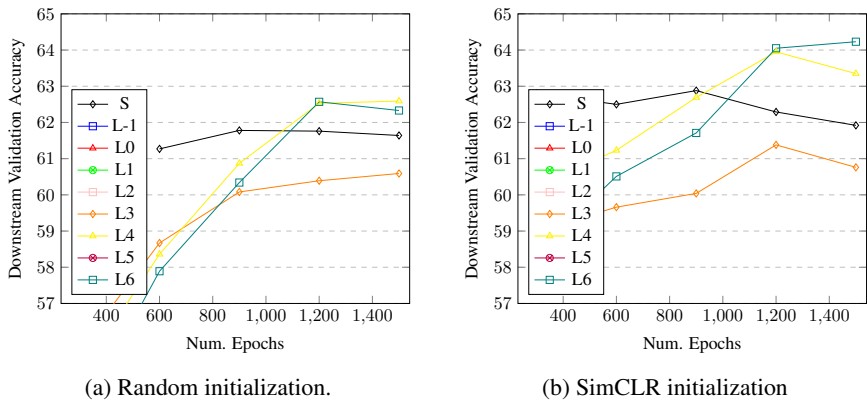

(a) Random initialization.                    (b) SimCLR initialization

Figure 46: Deep Augmentation with stop-gradient, only lower-to-higher augmentation pairs.

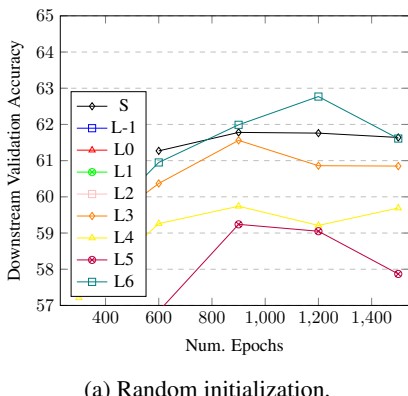

(a) Random initialization.

Figure 47: Deep Augmentation without stop-gradient, only lower-to-higher augmentation pairs.

## A.15    CIFAR100 Miscellaneous Experiments

We include some preliminary results on different aspects of Deep Augmentation that deserve further investigation.

In Figure 46, we include results of Deep Augmentation with stop-gradient where each pair consists of one sample that has only input-data augmentation and another sample that has input-data and higher-layer augmentation. I.e. we remove all the higher-to-higher and lower-to-lower pairs. We see that for Layer 4 and 6 the performance does not change substantially, but for Layer 3 performance degrades substantially.

In Figure 47, we include results of Deep Augmentation without stop-gradient where each pair consists of one sample that has only input-data augmentation and another sample that has input-data and higher-layer augmentation. I.e. we remove all the higher-to-higher and lower-to-lower pairs. We see that for the layers involved performance does not change substantially.

This suggests that lower-to-higher pairs are sufficient to make Deep Augmentation successful, but that certain layers are greatly helped by also including other lower-to-lower or higher-to-higher pairs.

In Figure 48, we include results of Deep Augmentation with stop-gradient and freezing layers before, but initialized with random weights instead of initialized with a pre-trained SimCLR model. We note that several layers are severely hurt by this compared to the SimCLR pre-trained model initialization.

In Figure 49, we include results of Deep Augmentation with stop-gradient and freezing layers before, but initialized with a model pre-trained with SimCLR and 20% dropout across all layers. We wanted to see if a model trained with high dropout everywhere was more helpful as a starting point for Deep Augmentation. Future work may investigate ways to optimally train a NN so that dropout serves as a useful higher transformation.

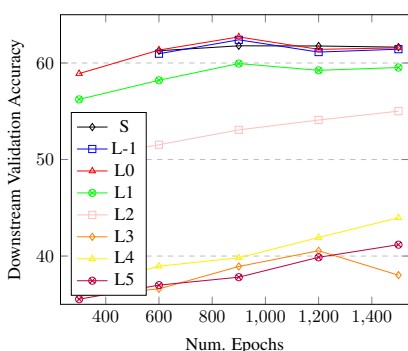

Figure 48: Deep Augmentation with stop-gradient and random initialization, freeze layers before.

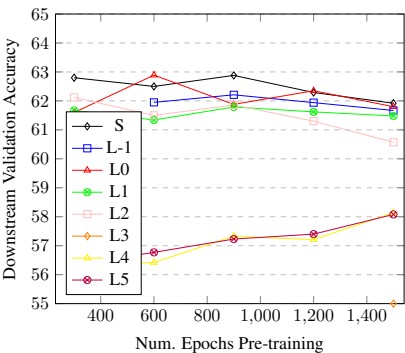

Figure 49: Deep Augmentation with stop-gradient and SimCLR-trained-with-20%-dropout initialization, freeze layers before.

# B  Sentence Embeddings

## B.1  Training Details

We used the training protocol of (Gao et al., 2021) with code released at link. Deep Augmentation at Layer 0 correspond to just after the first token-embeddings. Deep Augmentation at the subsequent layers was applied after each transformer layer in the code, with the last Layer 13 corresponding to the output latent vector.

## B.2  PCA Augmentation

In Figure 50, we include results with Deep Augmentation and subtracting the sixth largest principal component.

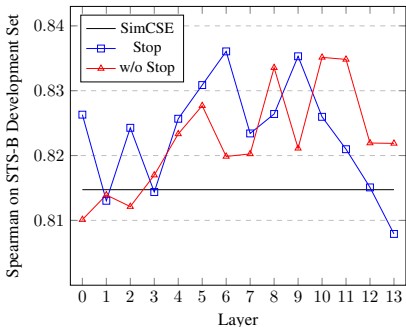

Figure 50: PCA: SimCSE vs. Deep Augmentation to 50% of the batch and subtracting the sixth largest principal component from that sample, with and without stop-gradient. "Stop": stop-gradient. Deep Augmentation outperforms SimCSE..

## B.3  Deep Augmentation and MLM only

In Figure 51, we include results with Deep Augmentation and MLM only, without contrastive learning. Deep Augmentation boosts performance substantially. This demonstrates that Deep Augmentation can boost performance in self-supervised learning settings beyond contrastive learning.

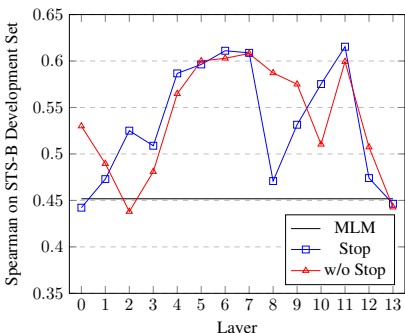

Figure 51: MLM vs. MLM with Deep Augmentation with and without stop-gradient, both without contrastive learning. "Stop": stop-gradient.

## B.4  Additional Results

In Figure 52, we include results of different dropout rates and hyper-parameter settings for using Deep Augmentation with SimCSE.

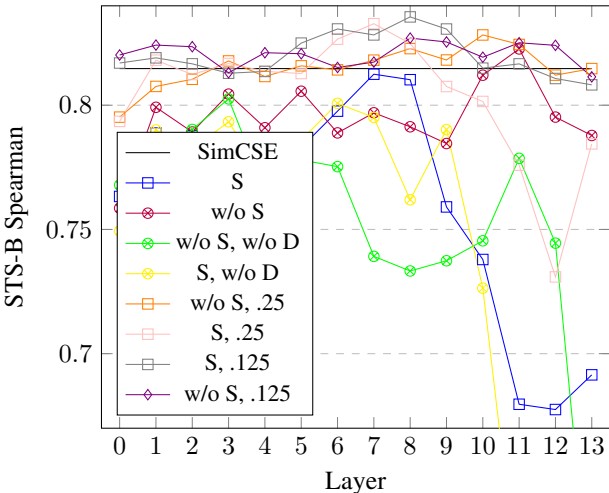

Figure 52: S: short for stop-gradient. D: short for default-dropout, referring to the 10% dropout (including attention-dropout) utilized by BERT and SimCSE. The decimal numbers refer to the Deep Augmentation drop out rate, and is .5 when unspecified.

# C    Graph Contrastive Learning

We follow the protocol and code of Zhu et al. (2021) that can be found at `https://github.com/PyGCL/PyGCL`. We pre-train for 1000 epochs and use the following data augmentations in GCL:

```
A.RandomChoice([
  A.RWSampling(num_seeds=1000,
    walk_length=10),
  A.NodeDropping(pn=0.1),
  A.FeatureMasking(pf=0.1),
  A.EdgeRemoving(pe=0.1)],
1)
```

We use f1mi (Micro-averaged F1 Score) as the evaluation metric. The `f1mi` metric computes the overall F1 score by aggregating the true positives (TP), false positives (FP), and false negatives (FN) across all classes. It is defined as:

$$\text{Precision}_{\text{micro}} = \frac{\sum_i \text{TP}_i}{\sum_i (\text{TP}_i + \text{FP}_i)},$$

$$\text{Recall}_{\text{micro}} = \frac{\sum_i \text{TP}_i}{\sum_i (\text{TP}_i + \text{FN}_i)},$$

$$F1_{\text{micro}} = 2 \cdot \frac{\text{Precision}_{\text{micro}} \cdot \text{Recall}_{\text{micro}}}{\text{Precision}_{\text{micro}} + \text{Recall}_{\text{micro}}}.$$

This metric is particularly useful for evaluating performance in imbalanced multi-class or multi-label classification tasks.

For ablation study with standard deviations, see Table 6.

## C.1    Dropout Rate Ablation

In Figure 53, we present an ablation study on various dropout rates. Note that the results reported are evaluated on the test sets. For Table 6, we report test results corresponding to the hyperparameters achieving the highest validation accuracy.

Table 6: Contrastive Learning on Graphs with GNNs. GCL (Graph Contrastive Learning) versus GCL with Deep Augmentation, dropout, PCA, and with and without stop-gradient.

| Model | COLLAB | IMDB-Multi | NCI1 | PROTEINS |
|---|---|---|---|---|
| GCL | 72.40±0.6 | 53.33±1.1 | 73.97±1.6 | 72.32±1.5 |
| GCL+DeepAug Drop L0 w/ S | 70.93±2.3 | 56.44±3.0 | 73.07±0.7 | 72.92±2.9 |
| GCL+DeepAug Drop L2 w/ S | 70.33±1.5 | 54.00±2.8 | 72.34±0.6 | 71.73±2.3 |
| GCL+DeepAug Drop L4 w/ S | 71.00±1.8 | 52.44±5.1 | 73.32±1.5 | 72.62±1.1 |
| GCL+DeepAug Drop L6 w/ S | **73.80±1.3** | 52.89±4.2 | **75.83±1.0** | 73.21±1.5 |
| GCL+DeepAug Drop L0 w/o S | 71.87±2.7 | **56.89±2.2** | 75.51±1.7 | 73.51±2.6 |
| GCL+DeepAug Drop L2 w/o S | 70.40±2.0 | 52.44±3.9 | 73.32±2.5 | **73.81±2.1** |
| GCL+DeepAug Drop L4 w/o S | 70.93±1.6 | 53.56±3.0 | 75.67±3.3 | 72.32±1.9 |
| GCL+DeepAug Drop L6 w/o S | 70.87±1.1 | 52.00±2.7 | 74.61±1.1 | **73.81±2.3** |
| GCL+DeepAug PCA L0 w/ S | 71.2±1.3 | **54.44±0.8** | 73.4±1.9 | **74.4±2.9** |
| GCL+DeepAug PCA L2 w/ S | 70.53±3.0 | **54.44±2.7** | 73.48±2.4 | 73.21±1.5 |
| GCL+DeepAug PCA L4 w/ S | 70.73±1.4 | 51.11±5.1 | 74.37±0.8 | 72.92±1.1 |
| GCL+DeepAug PCA L6 w/ S | **72.0±0.4** | 50.22±2.2 | **75.59±0.1** | 73.51±0.8 |
| GCL+DeepAug PCA L0 w/o S | 68.13±2.4 | 52.89±3.8 | 74.78±0.6 | **74.4±1.7** |
| GCL+DeepAug PCA L2 w/o S | 70.53±1.3 | 54.0±2.0 | 74.45±0.5 | 72.62±2.3 |
| GCL+DeepAug PCA L4 w/o S | 71.93±0.8 | 54.22±4.9 | 74.21±0.9 | 72.92±2.3 |
| GCL+DeepAug PCA L6 w/o S | 70.87±1.5 | 53.11±0.3 | 75.18±1.0 | 72.02±0.4 |

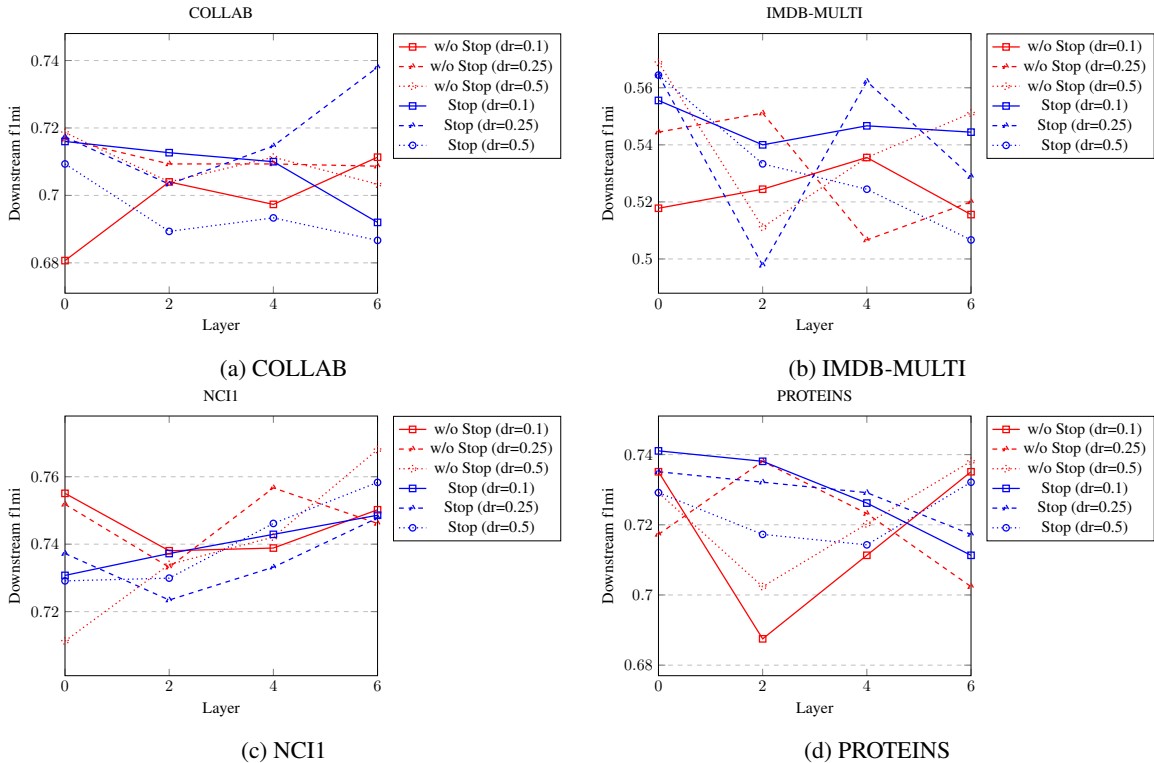

(a) COLLAB

(b) IMDB-MULTI

(c) NCI1

(d) PROTEINS

Figure 53: Ablation plots for dropout rates across datasets. Accuracy on test set.

# D  Alignment and Uniformity

First, we reiterate the fundamental drawbacks of Alignment and Uniformity for our work. Alignment is defined either with respect to a set of augmentations (the original intent Wang & Isola (2020)) or with respect to embeddings from different datapoints within the same class (as in SimCSE Gao et al. (2021)).

This poses the following issues for our work: (i) We must be very carefully to compare the Alignment and Uniformity results of sentences with those of images. (ii) Since Deep Augmentation introduces new augmentations, for images, we select only the default data augmentations for the Alignment and Uniformity metrics to enable comparison between Deep Augmentation, baselines, and other settings.

In Table 7, we present the Alignment and Uniformity measures for supervised models on sentence embeddings.

Table 7: Comparison of alignment and uniformity metrics across different models of Supervised Setting on Sentence Embeddings. *: Example of collapse

| Model | Alignment | Uniformity |
|---|---|---|
| Random Init | 0.032 | -0.514 |
| Regular | 0.790 | -2.415 |
| Deep Augmentation L8 w/ stop | 0.734 | -2.318 |
| Deep Augmentation L10 w/o stop | 0.671 | -2.153 |
| Deep Augmentation L1 w/o stop (Fail)* | 0.002 | -0.019 |

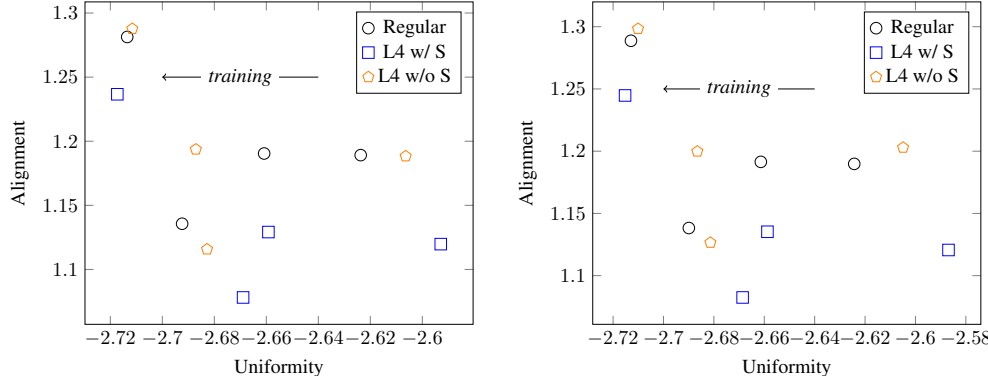

Figure 54: Alignment and Uniformity (lower is better) of Supervised model on SimCLR augmentations on CIFAR. Left: Test data. Right: Training data. Best performance on Alignment and Uniformity does not translate to best performance in supervised setting where the ground truth labels are known. "L" is short for Layer and "S" is short for stop-gradient.

# E  CKA Similarity Index Analysis

We include more complete results using CKA similarity index.

In Figure 55, we include results for several configurations for ResNet18 and CIFAR100. "Layer 4 without Stop" and "Layer 5 with Stop" do not perform well in their downstream performance and share the same increased co-adaptation between layers 4 and 5.

In Figure 57, we include results for several configurations for ResNet18 and CIFAR10. The same trends that were observed on CIFAR100 is also observed on CIFAR10.

Figure 58 displays results for various configurations on sentence embeddings and the STS-B development set. Deep Augmentation achieves optimal performance in the later co-adaptation region, with stop-gradient at its onset and without stop-gradient towards its conclusion.

In Figure 59, we also present the CKA similarity for supervised models. Additionally, Figure 60 shows the CKA similarity for a randomly initialized model. The supervised setting exhibits significantly less co-adaptation between

layers, particularly in the later layers. Although Deep Augmentation slightly decreases co-adaptation, this does not correlate with improved performance on the supervised task, suggesting that co-adaptation is less problematic for supervised learning compared to self-supervised learning. We include the "Deep Aug (Fail)" example to illustrate that training collapses, resulting in extremely low accuracy, are associated with very low co-adaptation, indicating that a nuanced level of co-adaptation is necessary to retain information from the data.

It is also worth noting that, since information cannot be created by a deterministic function (i.e., $I(X; Y) \geq I(X; f(Y))$), the reduction in co-adaptation through transformations likely corresponds to a removal of some information from the input data distribution. This suggests that reduced co-adaptation and less overfitting may be linked through the reduction of spurious information in the later layers of the neural network.

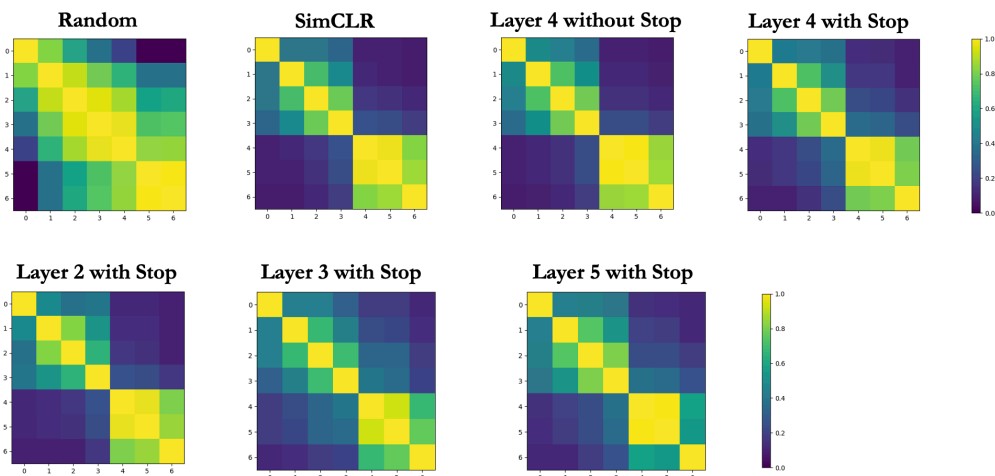

Figure 55: CKA similarity index of ResNet18 for different pre-training methods on CIFAR100.

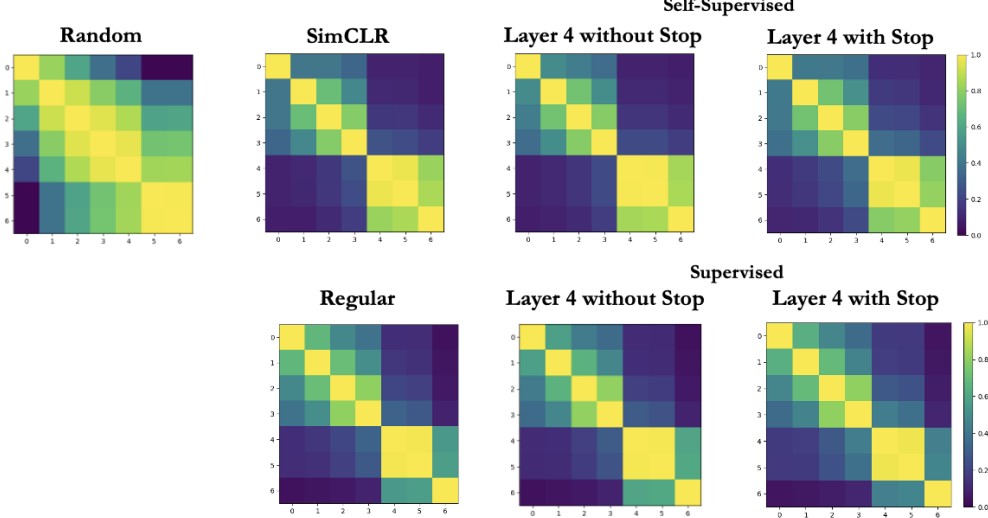

Figure 56: CKA similarity index of ResNet18 for and Deep Augmentation in contrastive learning (self-supervised) versus supervised learning settings on CIFAR100.

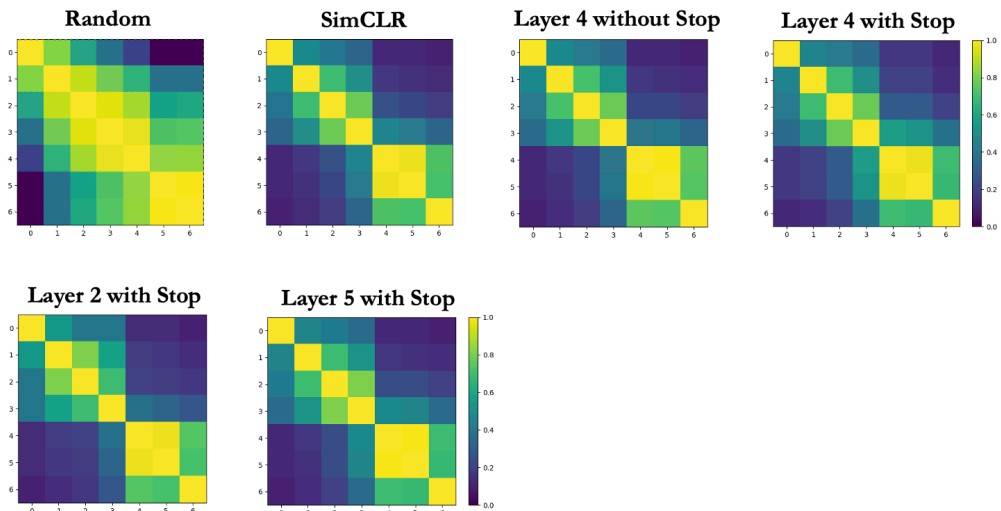

Figure 57: CKA similarity index of ResNet18 for different pre-training methods on CIFAR10.

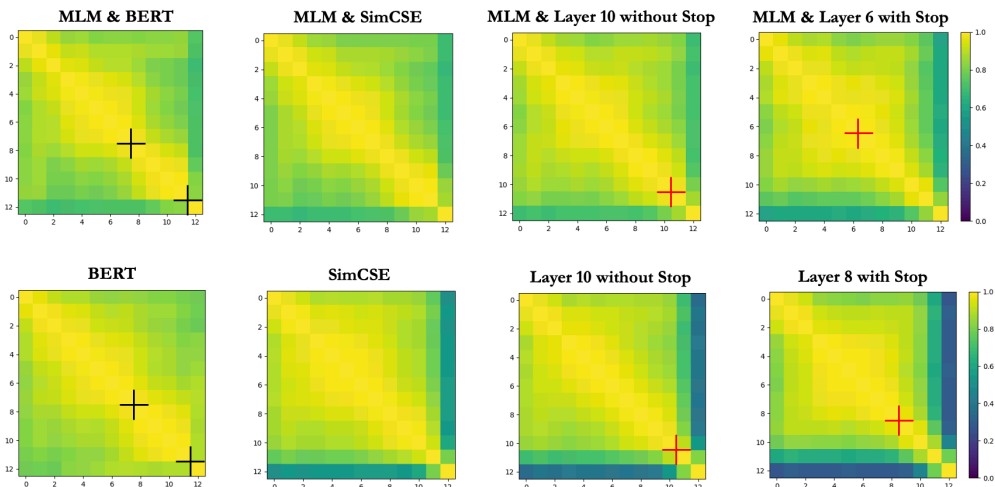

Figure 58: CKA similarity index for different methods trained to produce sentence embeddings. Black crosses indicate the start and end of co-adaptations stretch of layers in BERT, and red crosses indicate where the Deep Augmentation was applied. The layers at which Deep Augmentation performs the best are around the black crosses at the initialization "BERT".

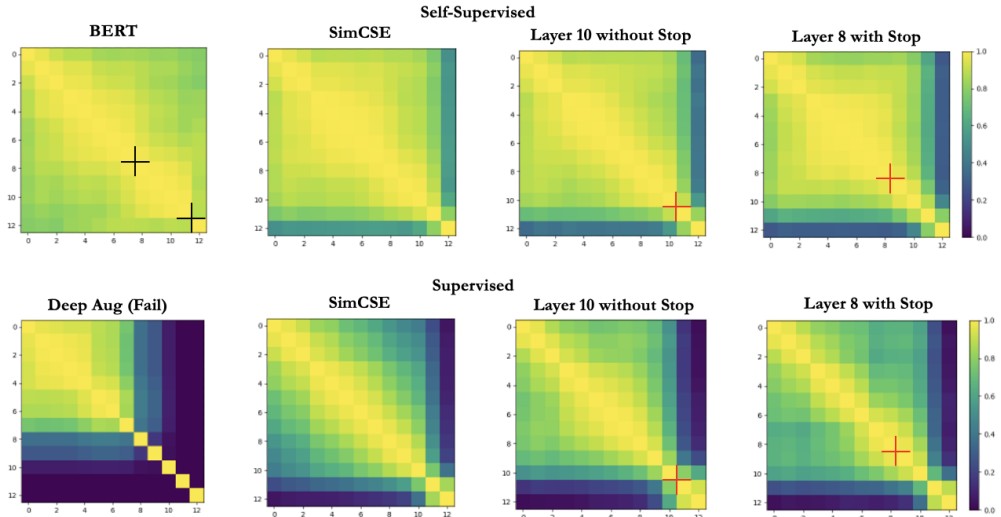

Figure 59: CKA similarity index for different methods trained to produce sentence embeddings. Black crosses indicate the start and end of co-adaptations stretch of layers in BERT, and red crosses indicate where the Deep Augmentation was applied. The layers at which Deep Augmentation performs the best are around the black crosses at the initialization "BERT". Upper row is with contrastive learning (self-supervised) setting and lower row is in supervised setting.

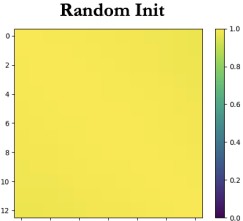

Figure 60: CKA similarity index for a random initialized model.

