# OpenReview forum: "Deep Augmentation: Dropout as Augmentation for Self-Supervised Learning"
_TMLR — Accepted by TMLR_

### Review · Reviewer_UJt2 · 2025-03-10

**Summary Of Contributions:**

This paper studies the role of dropout as a potential source of data augmentation in standard constrastive learning objectives used in self-supervised learning. Their work studies two main problems, (1) when is dropout useful as a source of data augmentation and (2) is dropout uniquely effective, when compared to alternative data augmentation approaches.

Regarding (1), the authors find that dropout (+ stop-gradient) helps performance in contrastive learning objectives but is generally harmful in supervised learning tasks. Furthermore, the stop-gradient operation seems crucial in SSL objectives, but is even more harmful for supervised learning. Both of these results are surprising and fairly interesting, although no explanation or further experimentation is provided.

The authors find that dropout hurts performance when applied uniformly across all layers, and is most beneficial when applied to target deeper layers in the network (when only applying to a single layer). The authors find that using PCA (to remove the component with the largest or 6th largest eigenvalue) as a data augmentation performs comparably to using dropout.

Experiments are performed on vision (with ResNets), NLP (with BERT-style models), and graph tasks (with graph convolutional networks).

**Audience:**

Yes

**Claims And Evidence:**

Yes

**Requested Changes:**

See the weakness above — I believe that moving ablations on dropout ratio into the main text is important and that adding ablations for the proposed method (i.e., dropout on individual targeted layers) on the other NLP and graph domains is quite important and are critical for my recommendation.

I also highly suggest adding the comparisons to other related data augmentation techniques (masking in the input layer or quantization) -- this would significantly strengthen the work in my opinion.

**Strengths And Weaknesses:**

## Strengths

* Experiments and results are demonstrated across a wide variety of tasks with different model architectures. Ablations show differences in performance when dropout is applied to specific layers in the network.

## Weaknesses

* Ablations regarding dropout ratio should be brought into the main text. To me, this seems like a crucial and important hyperparameter for this method. In fact, the amount of masking seems quite important for MAE-style approaches, so analyzing various different dropout rates should be more strongly highlighted in the paper.
* It seems that only Figure 34 contains an ablation on dropout rate for the proposed method (i.e., dropouts at an individual, specific layer). This should be expanded upon for other tasks (NLP or graph tasks).
* I feel that the drop in performance for supervised objectives could be explored and analyzed more in depth.
* I believe that the paper could benefit strongly from comparisons to similar data aumgnetation techniques, such as quantization [1] or masking (which is essentially dropout on the pixels for vision tasks) [2].

[1] Wu, et. al. Randomized Quantization: A Generic Augmentation for Data Agnostic Self-supervised Learning.
[2] Assran, et. al. Masked siamese networks for label-efficient learning.

---

> ### Author Response · Authors · 2025-04-08
> **Response**
>
> Thank you for your valuable feedback regarding the dropout ratio. We agree that this hyperparameter is crucial, especially given its importance in MAE-style approaches. Due to the extensive experimental results already included in the paper, some of the dropout ablations have been placed in the Appendix—for instance, the dropout rate ablation for NLP is shown in Figure 52. In our uploaded revision, we have also added ablation over dropout rates for the graph tasks in Appendix C.1, Figure 53.
>
> In our work, we are primarily focused on key comparisons such as dropout versus PCA, supervised versus unsupervised learning, and the effects of applying stop-gradient. Nonetheless, we recognize the importance of expanding these analyses.
>
> Additionally, we have compared our method to other input augmentation techniques. For NLP, we include comparisons with MLM (see Section 5.1 “Deep Augmentation and Masked Language Modeling (MLM)”), and for vision tasks, our experiments build upon standard augmentations (see Section 5.2 “Dropout is Not Sufficient Alone”). A similar approach is applied to GCL on graphs (see Section 5.3).
>
> We believe the current submission satisfies the acceptance criteria and empirical standard for TMLR. If there are particular experiments you require in the revision, please let us know and we will include them.

---

### Review · Reviewer_eAmY · 2025-03-27

**Summary Of Contributions:**

The paper investigates the idea of dropout as an augmentation strategy for self-supervised learning.  While this idea may seem not intuitive at first sight (at least it was not intuitive to me in the beginning), it is quite natural the way it is formulated and they way it is explored by the authors.  In particular, the authors consider multilayer artificial neural networks (ANNs) and consider the application of dropout at a particular layer of the ANN.  Thus, dropout can naturally be interpreted as an "augmentation" at the particular latent representation that would otherwise be communicated to the next layer in the architecture.  The authors call this approach "Deep Augmentation".  Among the findings of their work is that a similar result can be observed by performing a PCA-inspired "augmentation" at the same layer of the ANN.  Furthermore, the authors show experimentally that Deep Augmentation can outperform input-level augmentations.  Notably, the experiments cover a wide range of scenarios, allowing different ANN architectures and data modalities on which the ANNs are applied to.  These experiments include applications to sentence embeddings, standard computer vision datasets, and graph contrastive learning on COLLAB, IMDB-Multi, and PROTEINS.

**Audience:**

Yes

**Claims And Evidence:**

Yes

**Requested Changes:**

Please see above the listed weaknesses as suggestions for improvement of the paper.  Below I have some clarification questions.

**Questions for the authors**

**Q1**  Can you explain the notation that you use in the equation that is on display for PCA Augmentation (page 4)?  In general, can you say a few more words and explain how this augmentation method works?

**Q2**  In the tables of page 5 (e.g., Table 1, Table 2, etc.), what do the different numbers represent?  Shouldn't something be stated, at least in the caption?

**Strengths And Weaknesses:**

**Strengths**

**S1.**  I like the idea of looking at dropout as a data augmentation strategy.

**S2.**  The paper shows that on a wide range of scenarios the proposed method is applicable.


**Weaknesses**

**W1.**  I understand that the authors want to show wide applicability of their method in different ANN architectures and data modalities.  However, in my opinion this makes it a paper that is hard to read, in two different ways, for the majority of the people that do machine learning in one way or another.

On one hand, the flow seems a little bit awkward because the authors need to argue in different sections sequentially for the different ANN architectures and modalities that they have chosen to investigate.  This is really the nature of the paper, but somehow it feels a little bit odd.  On the other hand, personally, I cannot appreciate the diversity of experiments that are being conducted, and I am not sure about the importance of the results in two out of three cases that are studied.  In any case, for computer vision where I do have some familiarity the narrative makes sense.  So, I trust that the same is true for text and for graph neural networks.

In other words, something seems a little bit off with the flow of the paper, but honestly I do not have a good suggestion for the authors.  So, please take this with a grain of salt.

**W2.**  I think the paper would benefit if the authors would spend a few paragraphs describing the methods that they use for the different modalities.  Here I am following up on the fact that personally I am not familiar with graph neural networks or with NLP-related tasks and therefore all the important models are completely unknown to me (e.g., GCL).  Adding a short paragraph explaining the importance of each method selected and perhaps comparing it against other similar methods in their respective field, I believe would be a good thing that all the readers would benefit from, and those who know can safely skip reading.  Also, I know that some description is given in Sections 5.x but I feel this is too little and also a little bit too late.  Something like related work or related methods background section would be nice to have (perhaps even in the appendix).

---

> ### Author Response · Authors · 2025-04-08
> **Response**
>
> Thank you for your valuable feedback. We have uploaded a revision with the following updates:
>
> **PCA Augmentation Notation:**
> We have updated the description of the notation for PCA augmentation in Section 3.2. Specifically,
> 	 $$ \text{PC}\bigl(\{x_k : k \in I_b\}\bigr)[:,:p]$$
> denotes the $p$ largest eigenvectors of the covariance matrix computed on the centered mini-batch. We hope this clarification makes the method clearer.
>
> **Table Descriptions:**
> We have added, in the revision, further explanation regarding the values presented in the tables; see Section 4 for the relevant text. In Table 1, the values represent Spearman’s correlation (higher is better). In Tables 2 and 3, the numbers indicate classification accuracies (again, with higher values being preferable). Table 3 employs the f1mi (Micro-averaged F1 Score) metric; additional details can be found in Appendix Section C.
>
> **Graph Contrastive Learning (GCL):**
> For GCL, we have clarified in Section 5.3 that the graph is augmented—using operations such as node and edge deletion—to create two distinct views. The model is then trained to identify pairs of views originating from the same graph.
>
> We trust that these revisions address your concerns and improve the overall clarity of our paper.

---

### Review · Reviewer_zJQJ · 2025-03-31

**Summary Of Contributions:**

The article studies how to use dropout as a data augmentation strategy
in the framework of contrast learning in self-supervised learning.
Focusing on the questions on when this idea will work,
the effect of the choice of the layer in a neural network is studied.
Based on extensive experiments, it is found that when using a deep layer.
the performance of classification problems could be improved. In particular,
when dropout is combined with stop-gradient.
The effect of using PCA instead of dropout is also studied, reaching a similar conclusion.

**Audience:**

Yes

**Claims And Evidence:**

No

**Requested Changes:**

- Clarify the PCA augmentation method. What is the variable p in the definition of A_P?
Why there are no eigenvalues involved in the subtraction of each sample x_i?
Does the computation relies on the mini-batch I_b and how sensitive is the choice of batch size (if small)?

- It is unclear what the dropout augmentation means, during the training and test. How should one interpret the variable z_i^j in the input?

- The idea of stop gradient is not so clear. Do you use random initial weights for the layers before l if stop gradient is applied? It is a bit surprising that if these initial weights were random, one could still achieve good classification performance, as well as desired invariances.

- Add std in Table 2 and Table 3 as in Table 1.

- Clarify how the Compute is obtained from the cpu and memory usage, in which ratio?

- The supervised learning results seem to be a bit of surprise. You mentioned that "Deep Augmentation reduces performance in
the supervised setting - especially in higher layers". This is contradictory to the original dropout article of Hinton
where it is found that dropout can avoid significant overfitting.

- Clarify which dropout rate is used in Figure 4.

- ImageNet-100 dataset is not so meaningful to compare with existing results. Is it possible to include results on the standard ImageNet (1000 classes) dataset?

- It is not so clear what are the ground-truth invariances mean in Section 6.

**Strengths And Weaknesses:**

The article studies a very interesting question which could result in
improved empirical performance of existing contrast learning models.
The study is quite complete in terms of the types of datasets such as text, image, and graphs.
The study is complementary to some existing results in the literature of data augmentation,
which is based on changing input images.

The writing of the methodology could be made clearer, to help me to understand the main contributions.
The results could be strengthened by including error bars, as well as positioning with known results.

---

> ### Author Response · Authors · 2025-04-08
> **Response**
>
> Thank you for your valuable feedback.
>
> **Clarification of the PCA Augmentation Method:**
> In our approach, $p$ denotes the number of eigenvalues used in the subtraction for each sample $x_i$, and this computation indeed relies on the mini-batch $I_b$. Although the eigenvalues themselves are not directly subtracted, their magnitudes are used to select the top eigenvectors. Our experiments suggest that the method is not highly sensitive to the mini-batch size, likely because the noise introduced helps generate a wider variety of augmentations. We have uploaded a revision with a clearer explanation of the PCA augmentation.  We have uploaded a revision of the paper; text clarifying the points above can be found in Section 3.2.
>
> **Dropout Augmentation Explanation:**
> The variable $z_i^j$ represents the noise injected into each batch example (and into each sample of a pair), which results in distinct dropout masks for each instance. At test time, no noise is applied, meaning that dropout augmentation is used only during training. We have added further clarification on this in Section 3.2 our revision.
>
> **Stop-Gradient Implementation:**
> Stop-gradient is applied to only 50% of the batch so that the remaining samples update all weights. This ensures that every weight is updated over the course of training. Please refer to “Partial Batch Sampling” in Section 3.2 for additional details.
>
> **Standard Deviation in Tables:**
> In the current revision, we added standard deviations for the graph learning results in Table 3 and (they are also included in Table 6 in Appendix Section C). In preparing the camera-ready version, we will strive to include standard deviation values also in Table 2, as we already have in Table 1 and 3.
>
> Note that analogous experiments on CIFAR-10 in Appendix Section A.12 demonstrate equivalent results. Although our computational resources are limited and it might take longer than the rebuttal period, we will ensure these statistics are included in the camera-ready version if required.
>
> **Compute and Memory Savings:**
> The compute savings arise from reductions in both GPU time and memory usage. As detailed in Section 4 (“Compute & Memory Savings”), applying stop-gradient leads to roughly a 4x reduction in compute time and a 3x reduction in memory usage for the affected layers. We have provided additional clarification on this in Section 4 (“Compute & Memory Savings”) of our revision.
>
> **Supervised Learning Results and Dropout Rates:**
> The observation that Deep Augmentation reduces performance in the supervised setting—especially in higher layers—is indeed surprising. It is important to note that the dropout rates used are high and that the networks are smaller than those typically encountered, which may contribute to this result. For Figure 4, we applied dropout rates of 0.5, 0.25, and 0.125. Please see Section 5.4 of the latest revision for discussion.
>
> **ImageNet-100 Comparison:**
> While comparing with the ImageNet-100 dataset may not be ideal, we currently lack the computational resources to include experiments on the standard ImageNet (1000 classes) dataset. The numerous experiments in the current paper, on many datasets and modalities, help bolster confidence in our discussion.
>
> **Ground-Truth Invariances:**
> Ground-truth invariances refer to the natural variations within each ground-truth class—that is, the intra-class differences. For example, in MNIST supervised training, the model is exposed to the full range of variations for each digit because the training procedure leverages the ground-truth labels to capture these intra-class differences. We have clarified this point in the Analysis Section (Section 6) of our revision.

---

> > ### Comment · Reviewer_zJQJ · 2025-04-15
> > **positionning**
> >
> > I thank the authors for their answers. There are still two main points that I have:
> > - The PCA methodology does not fully make sense to me. If the PC vectors have unit norms, that the subtraction x_i will depend on the norm of x_i. If the norm of x_i is huge, then the subtraction would be negligible. In this situation, PCA would behave as if there were no PCA performed, therefore to claim that it can behave as Dropout is equivalent to say that Dropout is not helpful as well when the norm of x_i is large. I suggest the authors to make a further clarification on the average norms of x_i in your numerical results, in particular if there is any batch normalization or not. Otherwise, a citation of some related works on this type of PCA methods would be needed.
> >
> > - Regarding the supervised learning results (e.g. Fig 4), I think that a remark is at least needed to discuss your results with respect to
> > https://jmlr.org/papers/v15/srivastava14a.html, which shows the advantage of dropout in supervised learning problems for vision, speech recognition, document classification and computational biology. It is not clear to me what is the reason why you have obtained contradictory conclusions. If the reason is unclear, I would suggest the authors to reproduce some of the results in this article (srivastava14a), and then to compare with contrastive learning to see if you still have the same conclusion. This would be complementary to your current results.
> >
> > Fig 4 seems to contain some error in the caption, as I do not know what which curve to look for "dropout rates .5, .25, .125".

---

> > > ### Author Response · Authors · 2025-05-03
> > >
> > > Thank you for the constructive follow-up.
> > >
> > > ---
> > >
> > > ### 1. Clarifying the PCA-based augmentation
> > >
> > > **Issue raised**
> > > > “If the norm of $x_i$ is huge, then the subtraction would be negligible …”
> > >
> > > **Clarification**
> > > Thank you for catching that ambiguity. We realized the phrasing could be interpreted as a mere subtraction; the revised Section 3.2 now states the operation explicitly as an orthogonal projection, consistent with the implementation.
> > > Given a mini-batch $I_b=\{1,\dots,K\}$ we first center the features
> > > $$
> > > \tilde x_k = x_k - \mu, \qquad
> > > \mu = \frac{1}{K}\sum_{k \in I_b} x_k ,
> > > $$
> > > Then we stack the centered features into the column matrix
> > > $ \widetilde{X} = [\tilde{x}_1, \dots, \tilde{x}_K] \in \mathbb{R}^{d \times K} $ and compute $\mathrm{SVD}(\widetilde{X}) = U\Sigma V^\top $.
> > > Let $V_p \in \mathbb{R}^{d\times p}$ be the matrix of the top-$p$ right singular vectors.
> > > Each sample is augmented as
> > > $$
> > > A_p(x_i) = \bigl(I - V_p V_p^{\top}\bigr) \tilde x_i + \mu .
> > > $$
> > > Because we project onto the orthogonal complement of the top-$p$ principal components after centering, the transformation never amplifies vector norms (the operator $I - V_p V_p^{\top}$ has spectral norm $\le 1$ because it is an orthogonal projector), yet it still removes a substantial component whenever the data genuinely varies along those high-variance directions; hence its effect does not become negligible simply because $\lVert x_i\rVert$ is large.
> > >
> > > ---
> > >
> > > ### 2. Apparent disagreement with Srivastava et al. (2014)
> > >
> > > **Issue raised**
> > > > “Srivastava (2014) shows dropout improves supervised learning, yet your Fig. 4 suggests the opposite.”
> > >
> > > **Reply**
> > >
> > > Note that, unlike Srivastava (2014), we apply dropout exclusively at a single layer. Our intention is not to dispute the overall effectiveness of dropout in supervised learning, as Srivastava demonstrates its utility under specific conditions such as particular parameter regimes, learning rates, and training durations. Rather, our observation highlights that Deep Augmentation (targeted dropout), which benefits self-supervised learning under certain regimes, can be counterproductive when applied similarly in supervised contexts. Investigating the conditions under which Deep Augmentation can positively impact supervised learning remains an intriguing area for future work. We have clarified this point in Section 5.4.
> > >
> > > ---
> > >
> > > ### 3. Caption in Fig. 4
> > >
> > > The plots show the best performance across dropout rates $0.5$, $0.25$, $0.125$. We have updated the caption in Figure 4 accordingly to clarify this detail.
> > >
> > > ---
> > >
> > > We hope these clarifications address the raised points and improve the manuscript.
> > > Thank you again for the insightful comments that helped us improve the paper.

---

### Decision · Action_Editor_MfJc · 2025-05-14

**Recommendation:** Accept with minor revision

**Comment:**

This paper presents a question, "Can dropout serve as data augmentation to enhance self-supervised learning?" Although naively adding dropout does not help much, the authors provide a remedy to explore how to enhance the performance. After several rounds of the revisions, the experiments have been much more complete, thus we accept this paper. Please incorporate the reviewers' feedback appropriately in the final version.

**Audience:**

Yes, this paper is targeted at the most majority of TMLR audience.

**Claims And Evidence:**

Yes, the authors provides a wide range of experiments with ablation studies.